# Improved genetic discovery and fine-mapping resolution through multivariate latent factor analysis of high-dimensional traits

## Graphical abstract

## Authors

Feng Zhou, William J. Astle,
Adam S. Butterworth, Jennifer L. Asimit

## Correspondence

jennifer.asimit@mrc-bsu.cam.ac.uk

## In brief

Zhou et al. introduce flashfmZero for analyzing many traits together by considering relationships between their underlying latent factors. Applied to 99 blood cell traits, it improves the detection of genetic signals and provides more precise results than traditional methods. flashfmZero offers insights into the genetic basis of complex health traits.

## Highlights

- Latent factors cluster traits that have common underlying biological mechanisms

- Latent factor GWASs could be estimated from observed trait GWAS summary statistics

- Latent factor methods boost power to detect and fine-map signals for many traits

- flashfmZero improves fine-mapping resolution by leveraging multiple latent factors

 Zhou et al., 2025, Cell Genomics 5, 100847
May 14, 2025 © 2025 The Author(s). Published by Elsevier Inc.

# Cell Genomics

CellPress

## Article

# Improved genetic discovery and fine-mapping resolution through multivariate latent factor analysis of high-dimensional traits

Feng Zhou,[1] William J. Astle,[1,2,3] Adam S. Butterworth,[3,4,5,6,7] and Jennifer L. Asimit[1,8,*]
[1]MRC Biostatistics Unit, University of Cambridge, Cambridge CB2 0SR, UK
[2]NHS Blood and Transplant, Cambridge CB2 0PT, UK
[3]NIHR Blood and Transplant Research Unit in Donor Health and Behaviour, University of Cambridge, Cambridge CB2 0BB, UK
[4]British Heart Foundation Cardiovascular Epidemiology Unit, Department of Public Health and Primary Care, University of Cambridge, Cambridge CB2 0BB, UK
[5]Victor Phillip Dahdaleh Heart and Lung Research Institute, University of Cambridge, Cambridge CB2 0BB, UK
[6]British Heart Foundation Centre of Research Excellence, University of Cambridge, Cambridge CB2 0QQ, UK
[7]Health Data Research UK Cambridge, Wellcome Genome Campus and University of Cambridge, Cambridge CB10 1SA, UK
[8]Lead contact
*Correspondence: jennifer.asimit@mrc-bsu.cam.ac.uk

## SUMMARY

Genome-wide association studies (GWASs) of high-dimensional traits, such as blood cell or metabolic traits, often use univariate approaches, ignoring trait relationships. Biological mechanisms generating variation in high-dimensional traits can be captured parsimoniously through a GWAS of latent factors. Here, we introduce flashfmZero, a zero-correlation latent-factor-based multi-trait fine-mapping approach. In an application to 25 latent factors derived from 99 blood cell traits in the INTERVAL cohort, we show that latent factor GWASs enable the detection of signals generating sub-threshold associations with several blood cell traits. The 99% credible sets (CS99) from flashfmZero were equal to or smaller in size than those from univariate fine-mapping of blood cell traits in 87% of our comparisons. In all cases univariate latent factor CS99 contained those from flashfmZero. Our latent factor approaches can be applied to GWAS summary statistics and will enhance power for the discovery and fine-mapping of associations for many traits.

## INTRODUCTION

Many genetic variants associated with disease risks or quantitative traits have been identified by genome-wide association studies (GWASs).[1] Many examples of pleiotropy exist among these findings, where a variant affects several traits, often by affecting a pathway upstream of multiple related traits.[2] When genetic variants affect a group of traits through a common pathway, methods that leverage the shared signal in the component of genetic variation common to all the traits, while accounting for residual correlation, are able to identify associated variants (multi-trait GWASs, e.g., MTAG[3]) and pinpoint causal variants (multi-trait fine-mapping, e.g., flashfm[4]) more powerfully than methods that analyze traits individually. Such approaches provide an efficient way to gain statistical power without increasing sample size.

A complete blood count (CBC) report is an example of a multivariate phenotype in which correlation between the component traits arises, in part because of a common dependence on variation in one or more biological processes. All types of blood cells derive from a common stem cell type, the hematopoietic stem cell (HSC), and different types of blood cells interact, for instance, in hemostasis and in immune re-

sponses. CBCs include measurements of hemoglobin concentrations and of blood concentrations of reticulocytes, mature red blood cells, platelets, and the different types of white blood cells. Additionally, they often contain measurements of the mean cell volumes of several cell types. GWASs of CBC traits have been conducted using samples of hundreds of thousands of participants, identifying hundreds of associations with genetic variants. Many of these associations are shared by biologically related traits. For example, genetic variants that increase mean platelet volume usually also reduce platelet count, presumably because the proportion of blood volume occupied by platelets is physiologically regulated.[5] The missense variant rs3184504 in *SH2B3*, which encodes lymphocyte adapter protein (LNK), is associated with traits measuring properties of reticulocytes, mature red cells, neutrophils, eosinophils, basophils, lymphocytes, and monocytes.[6] LNK encodes an adaptor protein that regulates cytokine signaling in HSCs and plays a crucial role in HSC self-renewal and the differentiation of all the major blood cell lineages.[7,8] Because CBCs typically contain at least two dozen traits measured simultaneously, many of which are genetically and biologically correlated, they provide an ideal testing ground for multi-trait association methods.

Genetic studies of high-dimensional phenotypes, such as profiles of gene expression, protein or metabolite levels, rely heavily on univariate analyses, partly because most multi-trait GWAS methods are limited computationally to a handful of traits. Consequently, such studies do not leverage the information in association signals shared across phenotypes. Some multi-trait GWAS methods use individual-level data to fit a multivariate linear model, jointly testing for association between a variant and each of several traits (e.g., GEMMA[9]). Summary-level (GWAS summary statistics) methods have the advantage that their computational efficiency does not depend on sample size. They can be broadly partitioned into methods that: (1) jointly model effect size estimates from several traits (e.g., MTAG[3]), or (2) reduce the dimension of the genome-wide joint distribution of the GWAS effect sizes from multiple traits through factor analysis (e.g., genomicSEM[10] or FactorGo[11]).

Rather than taking a dimension reduction approach to the distribution of effect sizes aggregated from multiple single-trait GWASs, we take a different perspective and use factor analysis to investigate the GWASs of latent factors that underlie the traits. Factor analysis captures the covariation between multiple traits by modeling them jointly as linear combinations of a set of common latent factors (plus independent error terms). Such latent factors can correspond to common sources of biological variation for which we can estimate GWAS summary statistics.

Current multi-trait fine-mapping methods that allow multiple causal variants are not scalable to high-dimensional traits. CAFEH[12] and mvSuSiE[13] are multi-trait extensions of SuSiE[14] fine-mapping: CAFEH assumes that traits are independent and it allows for missing trait measurements, while mvSuSiE models trait correlations and requires complete data. Flashfm[4] accounts for trait correlations and leverages information between traits in a Bayesian framework, allowing for missing trait measurements; the prior on the model space allows traits to have shared and distinct causal variants and upweights multi-trait models with shared causal variants. CAFEH and flashfm provide trait-specific posterior probabilities of causality for each trait, analogous to MTAG multi-trait GWASs.[3] In contrast, mvSuSiE outputs a posterior probability that each variant is causally associated with at least one trait and uses a second metric (the local false sign rate) to infer which the associated traits are.

We introduce flashfmZero, an extension of flashfm to jointly fine-map association signals from any number of latent factors by taking advantage of their zero-correlation from varimax rotation. flashfmZero rapidly fine-maps signals with multiple uncorrelated traits (or latent factors). We also show that latent factor GWAS summary statistics can be derived from observed trait GWAS summary statistics and a factor loading matrix from the trait correlation matrix. This widens use of the latent factor GWAS and flashfmZero to summary-level datasets, offering the flexibility to include individuals with incomplete trait measurements.

To illustrate the performance of a latent factor GWAS and single/multiple latent factor fine-mapping and to compare them, respectively, with a univariate GWAS and univariate fine-mapping of multiple traits, we focus on 99 blood cell traits measured in a subset of the INTERVAL cohort of UK blood donors[15–17] who have measurements for all traits. We interpret the results of our analyses in the context of fine-mapping results from UK

Biobank (UKBB) as part of a much larger study.[6] We show that fine-mapping of signals using latent factors has better resolution than fine-mapping of the measured traits, with further gains by flashfmZero. Using the INTERVAL study, we also apply flashfmZero to latent factor GWAS summary statistics separately computed from GWAS summary statistics of (1) 99 blood cell traits[15,16] and (2) 184 metabolic traits.[18]

# RESULTS

## Identification of latent factors underlying variation in blood cell traits

We used blood cell trait data from the extended CBC reports generated by Sysmex XN hematology analyzers in >45,000 generally healthy UK blood donors from the INTERVAL study[15–17] (Table S1). We applied factor analysis with the varimax rotation to data from 18,310 INTERVAL participants with complete data to identify groups of blood cell traits sharing common latent factors. Using Horn's parallel method, we selected a model including 25 statistically uncorrelated latent factors (Figure S1, STAR Methods).

We calculated scaled factor loadings $C_{ij}$ to quantify each latent factor's contribution to each trait, i.e., the proportion of variance in blood cell trait $i$ explained by latent factor $j$, relative to the total variance explained jointly by the 25 latent factors (Table S2; STAR Methods). We describe the principal effects of the latent factors on the blood cell traits in Table S3.

Generally, blood cell traits that receive high contributions from the same latent factor belong to the same broad blood cell type (Figure 1; Table S2). For example, latent factor ML4 primarily explains variation in reticulocyte traits, while ML8 and ML5 are specific to basophil and platelet traits, respectively. Increased ML10 corresponds to increased platelet count without affecting platelet volume or other platelet characteristics. The volume of blood occupied by platelets (Plateletcrit [PCT]) therefore goes up with ML10. Increased ML17 corresponds to more reactive lymphocytes, while increased ML23 corresponds to reduced average cell volume, increased average cellular complexity, increased variability in cellular complexity, and increased average RNA content of both lymphocytes and monocytes.

Typically, the highest contributing latent factor is shared by highly correlated traits (Table S4), but there is no one-to-one mapping between latent factors and blood cell traits. Importantly, multiple latent factors make major contributions (i.e., $C_{ij} > 20\%$) to variation in some blood cell traits. For example, latent factor ML2—which varies closely with the mass of hemoglobin per red blood cell (mean corpuscular hemoglobin [MCH])—is a major contributor to variation in multiple red blood cell traits. It has a 41% contribution to RBC-SFL-DW (red blood cell side fluorescence distribution width) and 96% to RBC-FSC (red blood cell forward scatter). On the other hand, although ML21—which principally affects the distribution width of the mass of hemoglobin in red cells (red blood cell forward scatter distribution width [RBC-FSC-DW])—has a 25% contribution to RBC-SFL-DW, it contributes very little to variation in RBC-FSC. Notably, traits from the same broad cell type do not necessarily have the same primary contributing factor. For instance, ML1—which varies closely with neutrophil count (NEUT#)—is

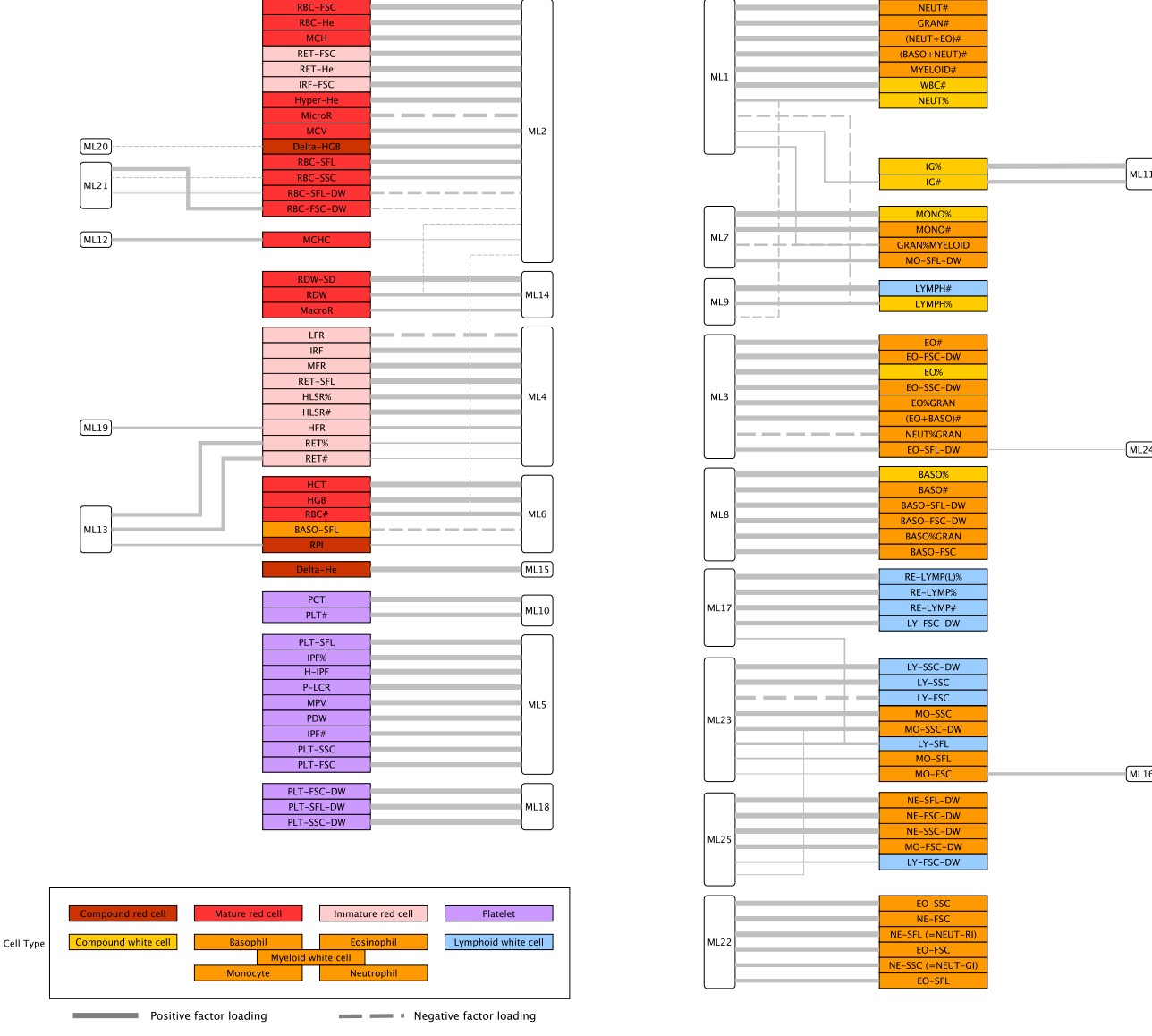

**Figure 1. Latent factors cluster blood cell traits grouped by broad cell type into groups with common underlying variance generating mechanisms**

Descriptions of the 99 traits, including abbreviations and full names, are given in Table S1 and their trait covariance matrix is given in Table S4. A line between a latent factor (open rounded rectangles, e.g., ML11) and trait (colored rectangles, e.g., IG% or IG#) indicates a latent factor contribution of at least 20% to the trait; line thickness is proportional to $C_{ij}$; solid lines for positive factor loadings and dashed lines for negative factor loadings. Traits are categorized by broad cell type according to the color code in the legend. Latent factors were calculated in 18,310 INTERVAL participants. Source data in Table S2, interpretations in Table S3.

the primary factor contributing to seven white cell traits that are direct functions of NEUT#, for six of which it contributes more than 83% of the total latent factor generated variance. On the other hand, ML22 makes the principal contribution to variation in average neutrophil volume (neutrophil forward scatter [NE-FSC], 91%), average neutrophil granularity/complexity (neutrophil side scatter [NE-SSC], 69%) and average neutrophil nucleic acid content (neutrophil side fluorescence [NE-SFL], 88%). ML22 also makes the principal contribution to average eosinophil volume (eosinophil forward scatter [EO-FSC], 70%), average eosinophil granularity/complexity (eosinophil side scat-

ter [EO-SSC], 93%), and average eosinophil nucleic acid content (eosinophil side fluorescence [EO-SFL], 63%), suggesting that the latent factor captures a biological mechanism common to the two types of granulocytes.

**A GWAS of latent factors identifies additional association signals over a blood cell trait GWAS**

For each of the 25 latent factors and 99 blood cell traits, we conducted a GWAS using individual-level trait measurements and genotype data from the 18,310 participants who contributed to the factor analysis. The process we used to obtain latent

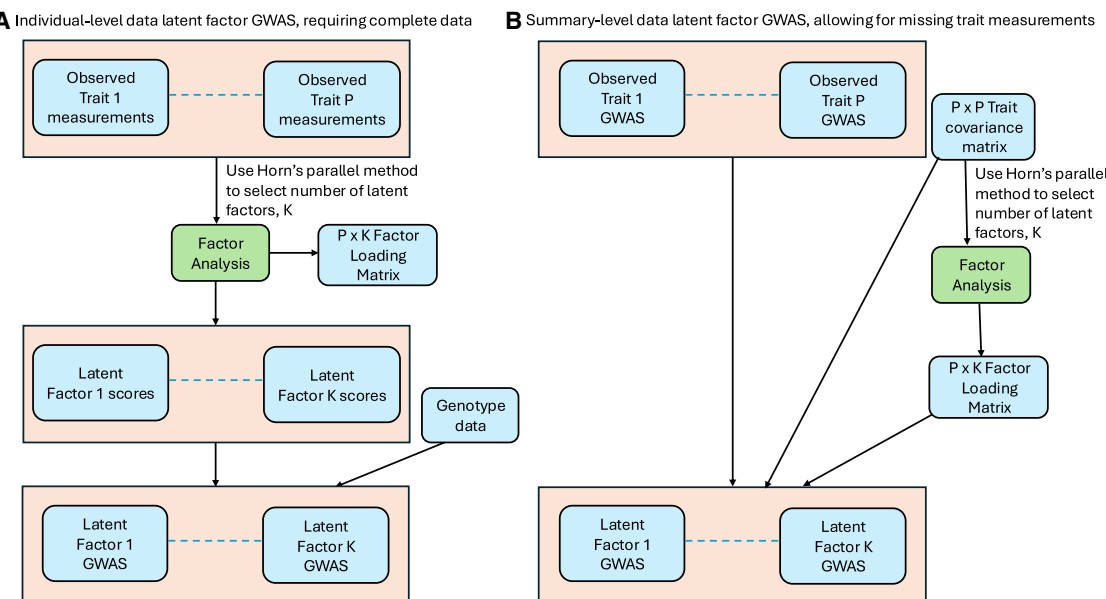

**A** Individual-level data latent factor GWAS, requiring complete data

**B** Summary-level data latent factor GWAS, allowing for missing trait measurements

**Figure 2. Flow diagram of latent factor GWAS summary statistics estimation**

Computation of latent factor GWAS summary statistics using (A) individual-level complete data: estimate latent factor scores from factor analysis of the observed traits then perform GWAS on each latent factor using genotype data. (B) Using observed trait GWAS summary statistics (allowing for missing trait measurements): estimate the factor loading matrix from the trait covariance matrix, for use with the observed trait GWAS summary statistics to calculate latent factor GWAS summary statistics.

factor GWAS summary statistics is illustrated in Figure 2A. Later, we relax the requirement for individual-level complete data and show that latent factor GWAS summary statistics can be computed from the observed trait GWAS summary statistics, allowing for missing trait measurements (STAR Methods; Figure 2).

To identify genetic association signals discovered by a latent factor GWAS but not a blood cell trait GWAS, we first selected the genome-wide significant (GWS) variants ($p < 5 \times 10^{-8}$) associated with each latent factor and clumped them by linkage disequilibrium (LD) ($r^2 > 0.6$). We then identified traits that receive a contribution of at least 1% from the latent factor and examined the trait $p$ values for association with the lead variants (those with the smallest $p$ value) from each of the latent factor clumps. Out of 3,399 lead variant associations, 3,036 had a GWS association with a connected blood cell trait, 211 had evidence for association at a suggestive significance threshold ($5 \times 10^{-8} < p < 1 \times 10^{-6}$) and 152 had no evidence at a suggestive threshold ($1 \times 10^{-6} < p$) at any of the connected blood cell traits (Table S5).

Next, to explore the signals across latent factors and blood cell traits, we formed LD clumps ($r^2 > 0.6$) of the set of variants with a GWS association with at least one of the 99 traits; separately, we formed clumps of the set of variants with a GWS association with at least one of the 25 latent factors. We assumed each clump to represent a distinct association signal and considered a signal identified by the blood cell traits to have been identified by the latent factors if any variant in the corresponding trait clump exhibited a GWS association with a latent factor. Symmetrically, we considered a latent factor signal to

have been identified by the blood cell traits if any variant in the corresponding clump exhibited a GWS association with a blood cell trait. As expected, we found that blood cell trait clumps that are significantly associated with multiple blood cell traits are more likely to be significantly associated with a latent factor (Armitage test for trend $p < 10^{-25}$); 67% (98/146) of the clumps associated with exactly two traits and 86% (102/119) of the variants associated with exactly three traits were also associated with a latent factor (Figure 3; Table S6). In contrast, 32% (96/301) of clumps associated with just one trait were also associated with a latent factor.

The advantage of factor analysis is illustrated by the 31 clumps that do not exhibit GWS associations with a blood cell trait, but that are significantly associated with a latent factor because they are moderately associated with several traits. For example, we found an association between ML8 and rs9310935 near *IL5RA* (per allele effect size estimate = −0.058SD, 95% confidence interval [−0.078SD, −0.037SD], $p = 3.3 \times 10^{-8}$) (Figure 4). *IL5RA* encodes the interleukin-5 receptor alpha subunit of a heterodimeric cytokine receptor found on the surface of eosinophils and basophils. Interleukin-5 signaling induces the differentiation and maturation of eosinophils in the bone marrow. Therapies specifically targeting this protein, such as benralizumab, are effective at blocking interleukin-5 signaling, reducing basophil and eosinophil counts through apoptosis, and therefore treating eosinophilic airway diseases such as severe eosinophilic asthma.

A significant association between rs9310935 and basophil count has been detected previously in a multi-ancestry meta-analysis,[19] but the signal did not reach the GWS threshold in either our single blood cell trait GWAS in INTERVAL or in a substantially

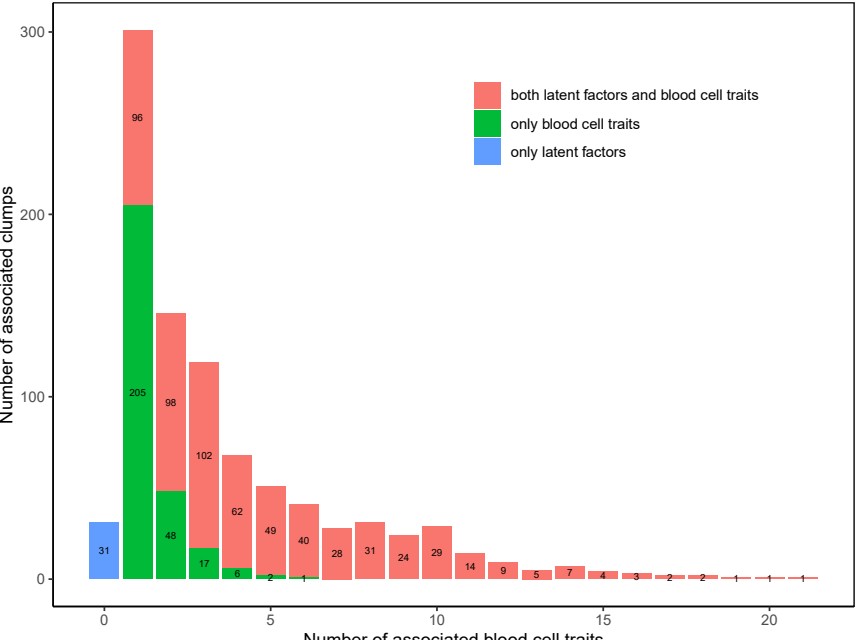

**Figure 3. Clumps associated with multiple blood cell traits are more likely to be associated with a latent factor**

This stacked barplot counts both blood-cell-trait-associated variant clumps and latent factor-associated variant clumps (MAF > 0.003, LD clumped $r^2$ > 0.6). The x axis indicates the number of distinct traits GWS associated with at least one variant in a clump. The y axis indicates frequency. Red bars count trait signals shared with latent factor(s), green bars count trait clumps not associated with latent factors, and the blue bar shows 31 clumps associated only with a latent factor. All GWAS data were based on 18,310 INTERVAL participants. Source data in Table S6B.

larger GWAS of European ancestry participants.[6,15] However, we did find moderate-to-weak evidence for association between rs9310935 and four basophil-related traits to which ML8 contributes variance (Figure 4; Table S5). Several other ML8-associated variants in the *IL5RA* region were associated with white blood cell traits in our analysis, as well as in previous studies[16] (Figures 4 and S2). rs9310935 remains significantly associated with ML8 when conditioning on the previously published lead variants, suggesting that its association signal is distinct from those previously identified (Figures 4 and S2).

Despite analyzing a relatively small sample (18,310 participants) compared with a previously published GWAS of blood cell traits, we identified 6 distinct GWS associations with the latent factors, after conditioning on the 3,559 lead variants from a recent large GWAS of complete blood cell traits[16] and Sysmex extended blood count traits.[15] These included the association between ML8 and rs9310935 in *IL5RA*. None of the 6 associated variants showed GWS evidence for association in our univariate analyses of the blood cell traits in the same participants (STAR Methods). In summary, among the 6 variants exhibiting novel associations, 4 of the variants were common (MAF > 0.01) with different likely causal genes and 2 were low frequency (0.003 < MAF < 0.01) from the same likely causal gene (Tables 1 and S7).

One of the common variants (rs6064377, MAF = 0.33) was associated with ML6 (per allele effect size estimate = −0.0609SD, 95% confidence interval [−0.083SD, −0.039SD], $p$ = 4.6 × $10^{-8}$), variation in which causes a change in hemoglobin (HGB) concentration and hematocrit (HCT) mediated by a change in red blood cell count (RBC#) while mean red corpuscular volume and MCH remain constant. The variant exhibited moderate evidence for association with HGB, HCT, and RBC#, with 1.5 × $10^{-7}$ < $p$ < 1.3 × $10^{-5}$ (Figure S3; Table S5). rs6064377 lies near

the gene *FAM210B*, which encodes the protein family with sequence similarity 210 member B, a mitochondrial membrane protein that is activated by GATA-1, a critical transcription factor for erythroid differentiation. *FAM210B* is thought to play a key role in regulating mitochondrial iron import to allow heme synthesis, thereby regulating erythropoiesis and iron transport, consistent with the associations seen with red blood cell traits.[20]

### Fine-mapping resolution gains are highest for joint latent factor fine-mapping

We considered 217 genomic regions that contain GWS associations with any blood cell trait at least 20% of the variance of which is explained by a latent factor with a signal in the same region (STAR Methods). Within each region, we applied JAMdynamic single-trait fine-mapping[21,22] to each latent factor (JAM latent factor) with a GWS association in the region and to all blood cell traits (JAM blood cell trait) that receive a contribution of at least 20% from these latent factors and have a GWS association in the region (Figure S4). We also applied multi-trait latent factor fine-mapping (flashfm latent factor) with flashfmZero (STAR Methods).

In previous simulation comparisons of multi-trait fine-mapping methods applied separately to European and African genetic ancestry individuals, flashfm had slightly higher power and lower false discovery rate (FDR) than mvSuSiE.[22] Moreover, mvSuSiE had noticeably higher FDR in the European sample, where there are longer LD blocks. As mvSuSiE requires complete data,[13] we applied it within the subset of 18,310 individuals and compared the results with those from flashfmZero in regions highlighted as having biologically likely causal variants. Our summary statistics version of flashfmZero can further improve fine-mapping resolution over methods requiring complete data because it is possible to include individuals who do not have measurements for all traits.

Let $CS99_{JAM-blood-cell-trait}$ be the size (number of variants) of a JAM blood cell trait CS99 (99% credible set), $CS99_{JAM-latent-factor}$ be the size of a JAM latent factor CS99, and $CS99_{flashfm-latent-factor}$ be the size of a flashfm latent factor CS99. We refer to a method

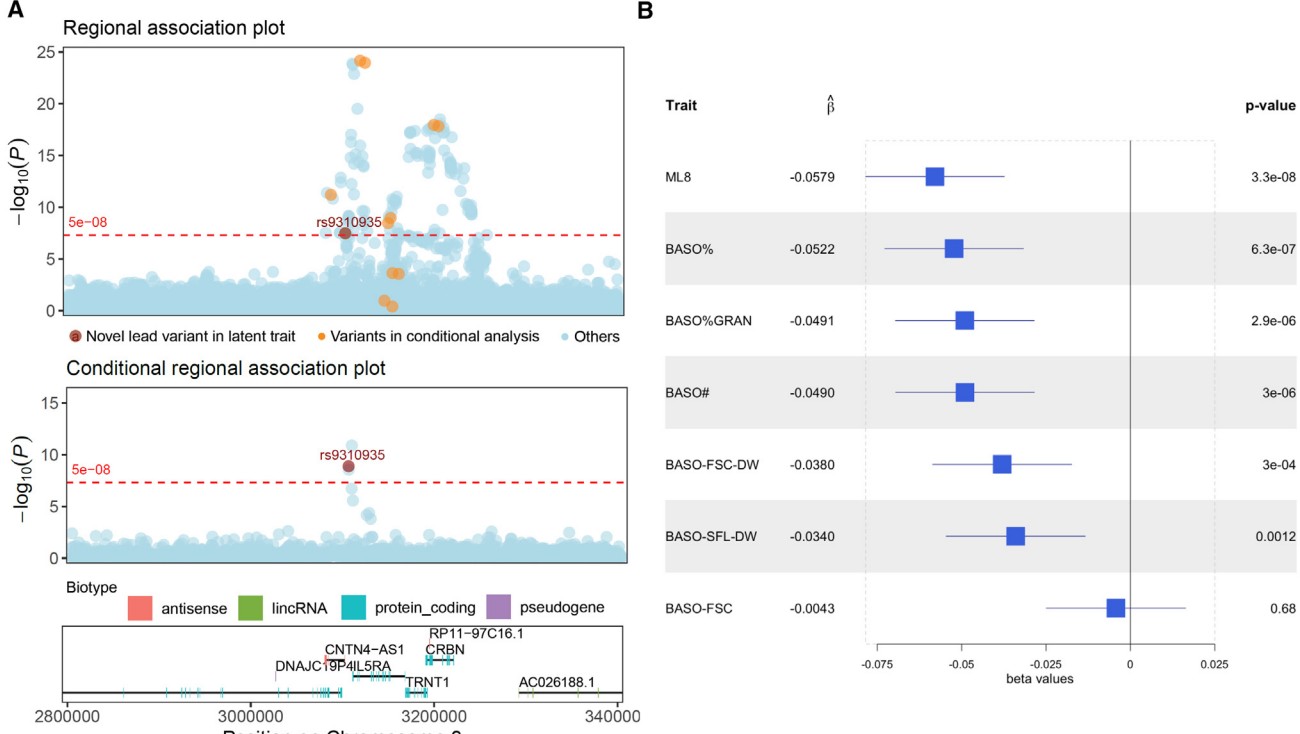

**Figure 4. Basophil-related latent factor ML8 is associated with rs9310935, which exhibits moderate evidence for association with multiple basophil-related traits**

(A) Regional association plot for ML8 (top panel), highlighting rs9310935 and conditional regional association plot for ML8 (middle panel), conditioned on 11 lead SNPs for basophil-related traits from previous publications, as detailed in Figure S2.

(B) Forest plot showing the rs9310935 effect size estimates and corresponding 95% confidence intervals for ML8 and its 6 linked basophil traits. All GWAS data were based on 18,310 INTERVAL participants. rs9310935 details in Table S7.

as having improved resolution over another method if it tends to construct smaller CS99s than the other method.

To compare results of the univariate blood cell trait fine-mapping to those of the single-trait and multi-trait (if at least two latent factors had a signal in the region) latent factor fine-mapping, we matched each blood cell trait to the latent factor that is the highest contributor to it, among latent factors that had a signal in the corresponding region (Figure S4). For comparisons between JAM latent factor and flashfm latent factor we matched by latent factor as these methods both return a CS99 for each latent factor. This resulted in 1,238 comparisons between $CS99_{JAM-blood-cell-trait}$ and $CS99_{JAM-latent-factor}$, 725 comparisons between $CS99_{JAM-blood-cell-trait}$ and $CS99_{flashfm-latent-factor}$, and 211 comparisons between $CS99_{JAM-latent-factor}$ and $CS99_{flashfm-latent-factor}$.

JAM latent factor has improved resolution over JAM blood cell trait. In 76% (937/1,238) of their comparisons, $CS99_{JAM-latent-factor} \leq CS99_{JAM-blood-cell-trait}$; in 58% (725/1,238) of them $CS99_{JAM-latent-factor} < CS99_{JAM-blood-cell-trait}$. flashfm latent factor further improves resolution over JAM blood cell trait, as $CS99_{flashfm-latent-factor} \leq CS99_{JAM-blood-cell-trait}$ in 86% (624/725) of their comparisons and $CS99_{flashfm-latent-factor} < CS99_{JAM-blood-cell-trait}$ in 71% (517/725) of them. When latent factors have no shared causal variants, as suggested

by no overlap in their CS99, flashfm latent factor and JAM latent factor give similar results (as previously illustrated for flashfm[4]). As flashfm methods have improved resolution over single-trait methods when traits share causal variant(s), we observe that $CS99_{flashfm-latent-factor} \leq CS99_{JAM-latent-factor}$ in 97% (205/211) of the comparisons and $CS99_{flashfm-latent-factor} < CS99_{JAM-latent-factor}$ in 45% (95/211) of them (Figure 5; Table S8).

We use the marginal posterior probability (MPP) of causality for a variant to assess accuracy of latent factor fine-mapping results. We compared the prioritized variants (MPP > 0.90) by our analyses with those prioritized (MPP > 0.95) in UKBB (approximately 500k individuals)[6]; we allowed a lower prioritization threshold in our comparatively smaller analysis, but most of our high-confidence variants do satisfy MPP > 0.95 (Table S8).

For this comparison we focused on 36 regions that met two conditions: (1) contained a prioritized variant by JAM blood cell trait, JAM latent factor, or flashfm latent factor in the INTERVAL analysis (MPP > 0.90), and by FINEMAP[23] in UKBB (MPP > 0.95),[6] and (2) the causal association in INTERVAL fine-mapping involved one of the 29 "classical" CBC traits analyzed in UKBB or with their linked latent factor(s) (STAR Methods). In 69% (25/36) of regions, at least one high-confidence variant identified by blood cell trait or latent factor

**Table 1. Latent factors allow the identification of associations that are not detectable in a blood cell trait GWAS of 18,310 individuals**

| Variant (chr:bp, rsID) | MAF | Likely causal gene | Biological support for likely causal candidate gene | Latent factor [cell type] (conditional $p$ value) | Conditional β (standard error) | Blood cell traits min $p$value (trait name) |
|---|---|---|---|---|---|---|
| 3:3101825 (rs9310935) | 0.4762 | *IL5RA* | *IL5RA* encodes the interleukin-5 receptor alpha subunit of a heterodimeric cytokine receptor found on the surface of eosinophils and basophils; therapies specifically targeting IL5-ra are effective at blocking interleukin-5 signaling and reducing basophil and eosinophil counts | ML8 [basophils] ($6.5 \times 10^{-9}$) | −0.0630 (0.0104) | $2.3 \times 10^{-7}$ (BASO%) |
| 5:40975803 (rs540446526) 5:41018263 (rs559314725) | 0.0036 0.0031 | *C7* | *C7* involves complement component 7, part of the terminal complement pathway of the innate immune system, which can activate platelets; patients with C7 deficiency show abnormal platelet aggregation, which can be corrected with addition of C7 | ML18 [platelet] ($1.0 \times 10^{-8}$) ML18 [platelet] ($2.2 \times 10^{-8}$) | −0.5132 (0.0901) −0.5402 (0.0971) | $7.0 \times 10^{-7}$ (PLT-FSC-DW) $4.8 \times 10^{-7}$ (PLT-FSC-DW) |
| 18:23024328 (rs72878322) | 0.3587 | *ZNF521* | zinc finger protein 521 (ZNF521) is a C2H2-type zinc finger transcription factor, which has been shown to repress erythroid differentiation by inhibiting GATA-1 activity, and to block B-lymphoid differentiation in primary hematopoietic progenitors by antagonizing early B-cell factor 1 | ML8 [basophils] ($1.8 \times 10^{-8}$) | −0.0624 (0.0112) | $2.8 \times 10^{-7}$ (BASO%) |
| 20:54884793 (rs6064377) | 0.3277 | *FAM210B* (family with sequence similarity 210 member B) | *FAM210B* encodes a mitochondrial membrane protein which is activated by GATA-1, a critical transcription factor for erythroid differentiation; *FAM210B* is thought to play a key role in regulating mitochondrial iron import to allow heme synthesis, thereby regulating erythropoiesis and iron transport | ML6 [mature red blood cells] ($4.4 \times 10^{-8}$) | −0.0618 (0.0112) | $1.8 \times 10^{-7}$ (HCT) |
| 21:23426550 (rs117617749) | 0.0642 | AP000472.2 | unclear - coding gene desert | ML6 [mature red blood cells] ($4.1 \times 10^{-8}$) | 0.1172 (0.0214) | $2.9 \times 10^{-7}$ (RBC#) |

Conditional two-sided $p$ values are calculated from association analyses conditioning on previously identified lead variants from large GWAS of blood cell traits.[15,16] Further details in Table S7.

approaches matched those identified in UKBB (Table S8). In an additional 11% (4/36) of regions, the prioritized variants of latent factor approaches matched those identified in UKBB, but no variants were prioritized by the blood cell trait fine-mapping.

Among the 25 regions where fine-mapped variants from both blood cell trait and latent factor approaches matched those in UKBB we found 9 regions for which latent factor fine-mapping prioritized causal variants for more traits than by blood cell trait fine-mapping. For example, in a region containing *PIEZO1* (a gene with a primary role in blood vessel formation and vascular

structure[24]), there were four correlated variants ($r^2 > 0.8$) in the CS99 for the mature red cell trait mean corpuscular hemoglobin concentration (MCHC), of which rs861400 (16:88862343) and rs551118 (16:88856084) had the highest (0.69) and second highest (0.27) MPP, respectively (Figure S5). However, rs551118 had the highest MPP for the immature red cell traits reticulocyte count (RET#) (MPP = 0.94; CS99 size = 4), reticulocyte percentage (RET%) (MPP = 0.88; CS99 size = 8), and reticulocyte production index (RPI) (MPP = 0.56; CS99 size = 8). ML12 primarily contributes to MCHC, while ML13 is the primary contributor to

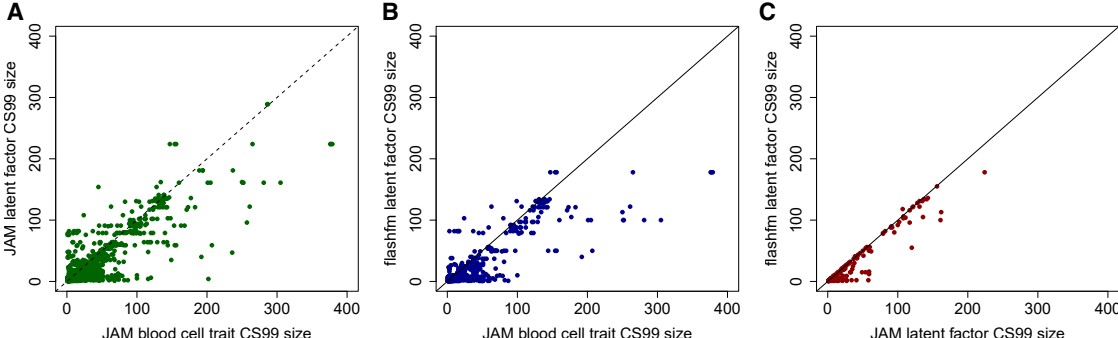

**Figure 5. Latent factor fine-mapping yields smaller 99% credible sets (CS99) than blood cell trait fine-mapping, with the largest resolution gain from joint fine-mapping of multiple latent factors**

There are (A) 1,238 comparisons between "JAM blood cell trait" and "JAM latent factor," (B) 725 comparisons between JAM blood cell trait and "flashfm latent factor," and (C) 211 comparisons between JAM latent factor and flashfm latent factor. CS99 sizes larger than 400 are not plotted; 11 CS99 for JAM blood cell trait, 10 CS99 for JAM latent factor, and 8 CS99 for flashfm latent factor. All GWAS data were based on 18,310 INTERVAL participants. Source data in Table S8.

RET#, RET%, and RPI (Figure S6). Our joint fine-mapping of the latent factors (flashfmZero) led to high-confidence that rs551118 is causally associated with both ML12 (MPP = 0.97) and ML13 (MPP = 0.96) (Figure S5). This suggests that rs551118 is the likely causal variant for MCHC, RET#, RET%, and RPI, which are linked with ML12 and ML13. The basophil-related traits linked with ML8 appear to have a distinct causal variant from rs551118, although we were unable to pinpoint it; rs904801 (16:88517105) has the highest MPP (0.338) for ML8 and has $r^2 = 0.002$ with rs551118.

After mvSuSiE identifies variant(s) with high posterior probability of causality for at least one trait (PIP), its local false sign rate (lfsr) is used to interpret which traits are associated with the variant(s); a recommended threshold[13] is *lfsr* < 0.01. Applying mvSuSiE to all traits observed to have a signal in the *PIEZO1* region also identified two potential causal variants: rs551118 (16:88856084) and rs904801 (16:88517105) with PIP values of 0.978 and 0.945, respectively. However, none of the traits had *lfsr* < 0.01 at either of these variants (Table S9). Weakening the threshold to *lfsr* < 0.05 suggests that all traits (related to red blood cells and basophils) have rs551118 as a causal variant, and none are impacted by rs904801. However, rs551118 is unlikely to be causal for basophil-related traits, which have $9.9 \times 10^{-4} < p < 0.038$, whereas rs904801 is GWS for these traits ($p < 2.4 \times 10^{-10}$) (Table S5). Moreover, mvSuSiE identified rs551118 as a causal variant for red blood cell traits, but not basophils, in an application to 16 blood cell traits within the European ancestry complete data subset of UKBB (248,980 individuals).[13] This suggests that large samples are needed for mvSuSiE to identify which traits have particular causal variants. For smaller studies (~20,000 individuals), mvSuSiE contributes to high-level identification of potential causal variants, but does not to a refined interpretation of the traits that are impacted by particular variants.

There were four regions in which only flashfmZero was able to prioritize variants that matched those identified in UKBB. In one of these regions, flashfmZero prioritized rs1175550 (1:3691528), an intronic variant of *SMIM1* (a regulator of red blood cell formation and the gene encoding the antigen underlying the Vel blood

group[25]), for three latent factors (ML4, ML12, ML14) that are all related to red blood cell traits (Figure 6). This result was validated by UKBB fine-mapping, which prioritized rs1175550 for nine red blood cell traits (e.g., HGB, RBC#, MCHC, RET#). It is also supported by published data showing rs1175550 to be an expression quantitative trait loci (eQTL) for *SMIM1* and a modulator of Vel blood group antigen expression.[26] The flashfmZero CS99s contained a single variant, a noticeable improvement over the blood cell trait CS99s—all containing rs1175550—with sizes 30–58 (Figure 6). rs1175550 had the highest MPP (0.24–0.59) in the univariate fine-mappings of the associations with HLSR# (high light scatter reticulocyte count), HLSR% (high light scatter reticulocyte percentage of red cells), MFR (medium fluorescent percentage of reticulocytes), and MCHC, and the second highest MPP (≈0.20) for IRF (immature fraction of reticulocytes) and LFR (low fluorescent percentage of reticulocytes), with the highest confidence variant (rs1175549 [1:3691727], MPP ≈ 0.26) being in high LD ($r^2 = 0.83$) with rs1175550. rs1175550 was second (MPP = 0.12) for RDW-SD (red cell distribution width—standard deviation), after rs7513053 (1:3709487; MPP = 0.44, $r^2 = 0.69$ with rs1175550). For RET-SFL (reticulocyte side fluorescence) rs1175550 ranked eighth (in a CS99 of 46 variants), with moderate LD ($r^2 = 0.47$) with the top variant (rs1175548 [1:3693032], MPP = 0.33). Single-trait latent factor fine-mapping improved resolution with CS99 sizes of 5–27; rs1175550 had the highest MPP for ML4 (MPP = 0.42) and ML12 (MPP = 0.92) and second-highest for ML14 (MPP = 0.18), although it has $r^2 = 0.69$ with the top variant, rs7513053 (1:3709487) (MPP = 0.33). flashfmZero further refined the CS99 values for the three latent factors, with rs1175550 having MPP > 0.99 (Figure 6).

The application of mvSuSiE to the *SMIM1* region gave results that agreed with flashfmZero, identifying rs1175550 (1:3691528) as a causal variant for at least one trait (PIP = 0.999). All traits with a signal in the region had *lfsr* < 0.01 (Table S9). This suggests that, in samples of this size (~18,000 individuals), mvSuSiE performs well when all traits share a causal variant, but that it has some difficulty identifying the traits impacted by causal variants when a region contains distinct causal variants for subsets of traits (as for *PIEZO1*).

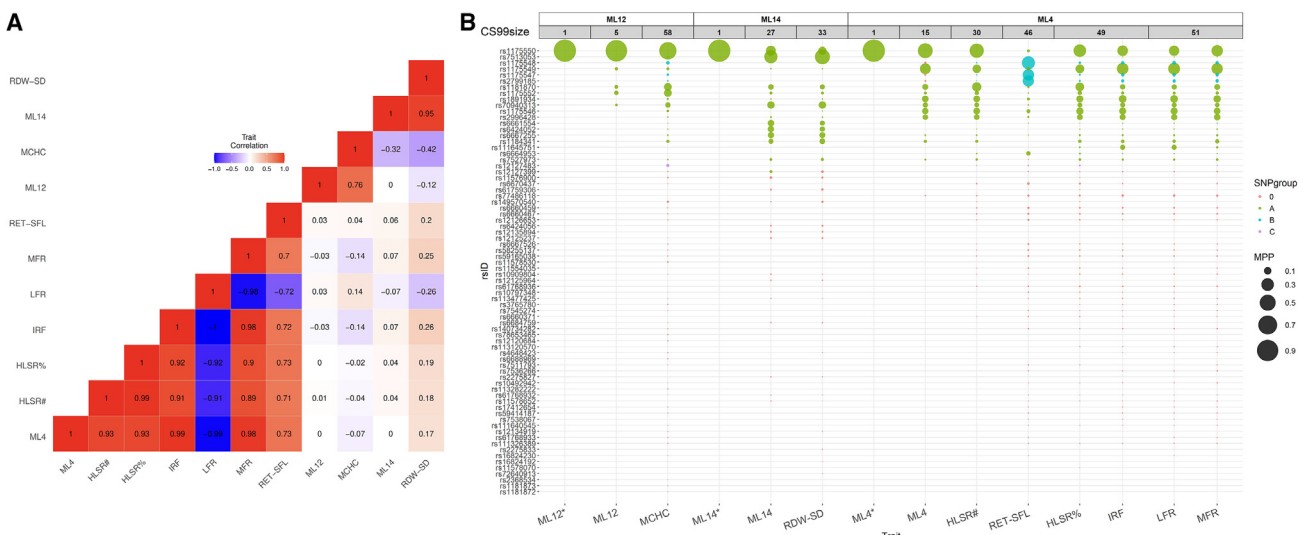

**Figure 6. Latent factors and fine-mapping in the *SMIM1* region**

(A) Correlations between latent factors and blood cell traits exhibiting genetic associations show high correlations between each latent factor and their linked traits, and among traits with a common latent factor. The three latent factors only contribute substantial variance to red blood cell traits—ML4 to six traits, ML12 to MCHC, and ML14 to RDW, as indicated by the correlation blocks.

(B) Comparison of 99% credible sets (CS99) for blood cell traits and latent factors. The multi-trait latent factor CS99s each contain one variant, refining the univariate latent factor CS99s, which in turn refine the univariate trait CS99s. Variants indicated by rsID on the y axis belong to at least one CS99, while columns correspond to CS99 from fine-mapping indicated on the x axis. Univariate latent factor CS99s are denoted by latent factor name (e.g., ML12), and those for multi-trait latent factors are appended with an asterisk (e.g., ML12*). Colored circles show variant membership in the CS99 for each trait, with colors (labeled by letters in the legend) indicating groups of variants (with marginal posterior probability [MPP] > 0.01, $r^2$ > 0.8) as calculated by the fine-mapping method; group "0" variants had MPP < 0.01 and were not assigned to a group. Each circle's area is proportional to the MPP that the variant is causally associated with the trait. Columns are grouped (open boxes top row) by latent factor contributions to traits, and within each group columns are ordered by CS99 size (e.g., ML12* has 1 variant, ML12 has 5, and MCHC has 58). All GWAS and trait correlations were based on 18,310 INTERVAL participants. Detailed results in Table S8.

Where latent factors are not biologically related, the flashfmZero and single-trait latent factor fine-mapping results are identical. For example, in a region containing *TMCC2*, we fine-mapped signals from 12 blood cell traits, of which 9 are platelet traits and 3 are basophil traits (Figure S7). No variants were prioritized by fine-mapping of individual blood cell traits; CS99 sizes were 16–52 for platelet traits and 18–19 for basophil traits (Figure S8). The 9 platelet traits are linked to ML5, which had a CS99 containing 8 variants. Likewise, the 3 basophil traits are linked to ML8, for which the CS99 was reduced to 12 variants. The CS99s of the basophil and platelet traits do not overlap, suggesting that they are unlikely to share any causal variants in this region, and hence flashfmZero results were identical to those from single-trait latent factor fine-mapping (Figure S8).

### Improved fine-mapping of signals in a latent factor GWAS estimated from summary statistics

We extended our method to use GWAS summary statistics, alleviating the need for complete individual-level data, and applied it to blood cell traits in up to 43,059 INTERVAL participants. We derived 25 latent factors from this larger sample (Figure S9), showing similar contributions to the 99 blood cell traits as factors derived from the subset of 18,310 participants (Figure S10). We selected 11 regions in which we had previously fine-mapped signals with blood cell traits and their latent factors using the complete data and repeated the fine-mapping using the summary statistics approach in the larger sample (Table S10).

In general, both approaches agreed on the selection of variants with the highest MPPs. As expected, in the larger sample, we identified association signals with additional traits and/or latent factors in some regions. We also observed that prioritized variants tend to have higher MPPs in the larger sample (Table S10) compared with the smaller sample (Table S8). We focus on the results of the comparison for *SMIM1* and *PIEZO1* (Table S9).

In the region containing *PIEZO1*, flashfmZero prioritized rs551118 (16:88856084) for ML12 and ML13, in agreement with the complete data analysis in the smaller sample. As for the complete data analysis, there was improved fine-mapping resolution for the latent factors over the observed traits. Further resolution gains were apparent with flashfmZero, including the refinement of the CS99s for ML12 and ML13 to a single causal variant (16:88856084). In the larger sample, four additional latent factors (ML10, ML14, ML15, ML20) and three additional traits (RDW-SD, MacroR, BASO-SFL-DW) showed signals in the region; RDW-SD and MacroR have contributions from ML14 and BASO-SFL-DW has contributions from ML8. The fine-mapping of ML14 in the larger sample also prioritized 16:88856084 with an MPP just below 0.99 (Tables S9 and S10).

Within the *SMIM1* region, compared with our previous analysis in the smaller sample, the summary-statistic-based analysis identified GWS associations with three additional red blood cell traits (HFR, RET#, RET%) and one additional latent factor (ML20). Single latent factor fine-mapping refined the CS99s of

the observed traits and flashfmZero further refined all CS99s to the single variant rs1175550 (1:3691528) (Tables S9 and S10). This agrees with our results from the smaller sample.

### Latent factors underlying variation in metabolic traits provide improved fine-mapping resolution

To demonstrate that our latent factor approach is generalizable beyond blood cell traits, we applied our framework to a set of 184 metabolic traits from the Nightingale Health nuclear magnetic resonance (NMR) assay platform (Table S11), also assayed in up to 40,849 participants from the INTERVAL study. We derived 21 latent factors from the trait variance-covariance structure (Figure S11; Table S12), which broadly corresponded to distinct biological categories but with varying specificity. For example, ML17 was specific to the ketone body acetone and ML8 was specific to conjugated linoleic acid (Table S13), while ML1-ML5 all ranked highly for a large number of lipid and lipoprotein parameters, reflecting the trait covariance structure (Table S14). We derived latent factor GWAS summary statistics from the factor loading matrix, trait covariance matrix, and NMR GWAS summary statistics.[18]

We fine-mapped associations in 11 1MB regions in which both latent factors and NMR traits had GWS association signals. Consistent with our blood cell trait analyses, we observed a general pattern that latent factor fine-mapping gave better resolution than fine-mapping of NMR traits, and flashfmZero gave better resolution than single-trait latent factor fine-mapping (Table S15). We identified previously reported causal variants in well-known metabolic genes. For example, ML2 (contributor to lipids in pro-atherogenic particles) prioritized p.R46L (rs11591147; 1:55505647), a loss-of-function missense variant in *PCSK9*. *PCSK9* encodes proprotein convertase subtilisin/kexin type 9, which degrades the low density lipoprotein (LDL) receptor and regulates levels of LDL-cholesterol in circulation. We also found that ML12 (contributor to omega-3 fatty acids) prioritized the missense variant 11:68562328 (rs2229738) in *CPT1A* in a single-variant CS99. *CPT1A* encodes carnitine palmitoyltransferase 1A, a rate-limiting enzyme in the fatty acid oxidation pathway.

We found that latent factor fine-mapping was able to dissect multiple signals in a region. For example, the region containing *TOMM40L* and *APOA2* had GWS-associated variants with 6 latent factors and with 12 observed traits, each of which had contributions of at least 20% from the 6 latent factors. Joint latent factor fine-mapping showed that there are at least 3 distinct signals in this region. The CS99 sizes for the observed traits ranged from 4 to 52 with the MPP of the top SNP varying from 0.10 to 0.81. Using latent factor fine-mapping we are able to prioritize one variant, 1:161619363 (rs10737488; MPP = 0.93) for association with ML13, which contributes to very low-density lipoprotein particle compositions. Joint latent factor fine-mapping prioritized the same variant for association with ML13 (MPP = 0.93), and prioritized 1:161623025 (rs61804164; MPP = 0.97) for association with ML10 (contributor to large and extra-large lipoprotein compositions). The variant 1:161194641 (rs4656292) has the largest MPP (but has MPP < 0.90) for association with small and medium high density lipoprotein (HDL) traits (with CS99 sizes of 4–9) and for association with the latent factors ML3 (contributor to small

and medium HDL traits, four variants in flashfmZero CS99) and ML14 (contributor to albumin concentration, four variants in flashfmZero CS99) (Table S15).

## DISCUSSION

Using blood cell traits and metabolic traits as examples of high-dimensional phenotypes with heterogeneous correlation structures, we demonstrated that, where multiple phenotypes capturing common biological variation have been measured, genetic association analysis of latent variables from factor analysis complements univariate analyses. This approach has two main advantages. Firstly, latent factors identify groups of measured traits influenced by common biological mechanisms, enabling inferences about groups of traits that share the same underlying factors. Secondly, a GWAS of latent factors can boost power to detect signals that may be missed in a GWAS of measured traits, when a variant exhibits only moderate evidence for association with each of multiple measured traits.

The example of *SMIM1* and red blood cell traits demonstrates that multi-trait fine-mapping of latent factors using flashfmZero can significantly improve resolution. This is because orthogonal latent factors may share causal variants if they capture aspects of a common biological process, even though they are, by mathematical definition, uncorrelated. However, we note that when the latent factors are not biologically related, it is less likely that they will share causal variants. In such instances, multi-trait latent factor fine-mapping will give similar results to univariate latent factor fine-mapping, although there will often be resolution gains over univariate fine-mapping of the measured traits.

We first investigated the analysis of a latent factor GWAS when individual-level data are available on the traits and genotypes. We used the complete data subset of study participants to calculate latent factor scores and performed GWASs with the latent factors. We also compared flashfmZero with mvSuSiE multi-trait fine-mapping,[13] which requires complete data. In smaller studies (~20,000 individuals), mvSuSiE could not dissect the relationships between variants and traits, but was able to do so in a substantially larger study[13] (>200,000 individuals). However, flashfmZero applied to the smaller study showed agreement with mvSuSiE applied to the larger study.

A latent factor GWAS using individual-level data is performed independently of an observed trait GWAS. However, we derived an approach to conducting a latent factor GWAS using only observed trait summary-level data, removing the complete data requirement. This also widens the scope of the latent factor approach to summary-level datasets freely available on-line, where the trait correlation may be estimated directly from individual-level trait data or using methods such as cross-trait LD Score regression.[27]

We demonstrated flashfmZero on uncorrelated latent factors derived from quantitative traits. Like flashfm,[4] we may apply our latent factor framework and flashfmZero to binary traits. Binary trait GWAS summary statistics from a linear model may be used directly in flashfm using the genetic correlation estimated by cross-trait LD Score regression,[27] while log odds-ratios from a logistic model should be converted to a linear approximation.[28] If binary traits have low/zero correlation, factor analysis is

inappropriate, but flashfmZero could be applied directly to any number of binary traits. For correlated binary traits, factor analysis using genetic correlation may estimate the number of latent factors underpinning multiple outcomes and the corresponding factor loadings. Latent factor GWAS summary statistics may then be calculated with the "latentGWAS" function in our flashfmZero package, followed by flashfmZero fine-mapping. Further work is needed to understand possible applications to rare diseases as to how well latent factors capture their variability.

Our analyses have illustrated the value of latent factor GWASs, with clear gains in fine-mapping, especially when signals from multiple latent traits are jointly fine-mapped. Further gains could be attained by incorporating functional annotations in the prior probabilities, an approach taken in PAINTOR,[29] PolyFun.[30] An extension of flashfm[31] that incorporates functional annotations was applied to fine-map glycemic trait genetic associations and has been shown to give significant improvements in resolution over annotation-agnostic flashfm and annotation-informed FINEMAP.[23]

Finally, flashfmZero has potential to inform shared therapeutics. For example, PIPE (pleiotropy informing prioritization and evaluation) uses pleiotropic evidence to prioritize and evaluate therapeutic targets by considering genetic variants identified by a cross-disease GWAS.[32] Its authors suggest that including flashfm-identified shared causal variants could give improvements. Integrating latent factors underlying many diseases or disease-related traits, and the improved prioritization of their shared causal variants via flashfmZero, may reveal additional insights to strengthen PIPE. Popular platforms like Priority index[33] and Open Targets Genetics[34] translate GWAS associations into drug target prediction, but they do not yet integrate pleiotropic evidence. However, both incorporate statistical co-localizations between trait GWAS signals and eQTL. Considering flashfmZero's gains in causal variant prioritization, it has high potential to identify shared causal variants underlying latent factors that explain multiple diseases and traits. Moreover, leveraging information between latent factors gives an alternative strategy to gain power that is crucial for smaller GWASs, as is common in eQTL studies and in under-represented populations. Integration of flashfmZero with Priority index and Open Targets Genetics could improve drug target identification and prioritization.

### Limitations of the study
Calculation of latent factor GWAS summary statistics from the observed trait GWAS summary statistics for a particular variant requires that summary statistics are available for all the observed traits. However, as the traits are measured in the same cohort, statistics for most variants are likely to be present for all traits.

Currently, flashfmZero assumes a single genetic ancestry group. We account for population structure by using GWAS summary statistics from a linear mixed-model (e.g., BOLT-LMM[35]). Extending to multi-ancestry studies would be useful, although not straightforward. Latent factor GWASs could be conducted within each genetic ancestry group. However, latent factors from each group must be equivalent for any analyses across groups, such as meta-analysis using MR-MEGA[36] or multi-

ancestry fine-mapping of latent factor signals via MGflashfm[22] or an extension of MeSuSiE.[37]

MGflashfm multi-group/ancestry multi-trait fine-mapping allows inclusion of genetic variants not present in all ancestry groups, identifying both shared and ancestry-specific causal variants. It uses flashfm to leverage information between traits within each group and performs joint analysis across groups using ancestry-specific LD panels. Future work includes adapting latent factor GWAS and flashfmZero to the MGflashfm framework for multi-group/ancestry multiple latent factor fine-mapping.

### RESOURCE AVAILABILITY

#### Lead contact
Further information and requests for resources should be directed to and will be fulfilled by the lead contact, Jennifer Asimit (jennifer.asimit@mrc-bsu.cam.ac.uk).

#### Materials availability
This study did not generate new unique reagents.

#### Data and code availability
Summary statistics from the latent factor GWAS are available from the GWAS catalog (https://www.ebi.ac.uk/gwas/) with accession numbers GCST90559243, GCST90559244, GCST90559245, GCST90559246, GCST90559247, GCST90559248, GCST90559249, GCST90559250, GCST90559251, GCST90559252, GCST90559253, GCST90559254, GCST90559255, GCST90559256, GCST90559257, GCST90559258, GCST90559259, GCST90559260, GCST90559261, GCST90559262, GCST90559263, GCST90559264, GCST90559265, GCST90559266, GCST90559267. Custom code for the INTERVAL blood cell trait analyses that use complete data are available at https://github.com/fz-cambridge/flashfmZero-INTERVAL-analysis and on Zenodo, https://doi.org/10.5281/zenodo.14992774.[38] Fully annotated scripts for the summary-statistics-based approach applied to the NMR traits of INTERVAL are available as articles at https://jennasimit.github.io/flashfmZero and on Zenodo, https://doi.org/10.5281/zenodo.13305579.[39] This same code was used for the summary-statistic-based analysis of the blood cell traits, but with minor changes.

FlashfmZero and our latent GWAS summary statistics estimation are freely available in our flashfmZero R package at https://jennasimit.github.io/flashfmZero and on Zenodo, https://doi.org/10.5281/zenodo.13305579.[39]

### ACKNOWLEDGMENTS

J.L.A. and F.Z. are supported by the UK Medical Research Council (MR/R021368/1 [to J.L.A.], MC_UU_00002/4). J.L.A. is also funded by the Isaac Newton Trust and Medical Research Foundation (MRF-DA-111). W.J.A. is supported by NHS Blood and Transplant. A.S.B. was supported by core funding from the British Heart Foundation (RG/18/13/33946, RG/F/23/110103), NIHR Cambridge Biomedical Research Centre (NIHR203312), BHF Chair Award (CH/12/2/29428), and by Health Data Research UK, which is funded by the UK Medical Research Council, Engineering and Physical Sciences Research Council, Economic and Social Research Council, Department of Health and Social Care (England), Chief Scientist Office of the Scottish Government Health and Social Care Directorates, Health and Social Care Research and Development Division (Welsh Government), Public Health Agency (Northern Ireland), British Heart Foundation, and the Wellcome Trust.

Participants in the INTERVAL randomized controlled trial were recruited with the active collaboration of NHS Blood and Transplant England (www.nhsbt.nhs.uk), which has supported field work and other elements of the trial. DNA extraction and genotyping were co-funded by the National Institute for Health and Care Research (NIHR), the NIHR BioResource (http://bioresource.nihr.ac.uk), and the NIHR Cambridge Biomedical Research Centre (BRC-1215-20014). The academic coordinating center for INTERVAL was supported by core funding from the: NIHR Blood and Transplant Research Unit (BTRU) in

Donor Health and Genomics (NIHR BTRU-2014-10024), NIHR BTRU in Donor Health and Behavior (NIHR203337), UK Medical Research Council (MR/L003120/1), British Heart Foundation (SP/09/002, RG/13/13/30194, RG/18/13/33946), NIHR Cambridge BRC (BRC-1215-20014, NIHR203312), and by Health Data Research UK. A complete list of the investigators and contributors to the INTERVAL trial is provided in Moore et al.[17] The academic coordinating center would like to thank blood donor center staff and blood donors for participating in the INTERVAL trial. We thank Parsa Akbari for making available blood cell traits from the INTERVAL study adjusted for technical variation. The views expressed are those of the authors and not necessarily those of the NIHR or the Department of Health and Social Care. For the purpose of Open Access, the authors have applied a CC BY public copyright license to any Author Accepted Manuscript version arising from this submission.

## AUTHOR CONTRIBUTIONS

J.L.A. conceptualized the study and developed the statistical methodology and related software. F.Z., W.J.A., and J.L.A. analyzed the data. F.Z. produced visualizations and contributed to the software implementation. A.S.B. and W.J.A. provided domain knowledge on metabolic and blood cell traits. J.L.A., A.S.B., and W.J.A. interpreted results and wrote the paper. A.S.B. provided the data. J.L.A., A.S.B., W.J.A., and F.Z. reviewed and approved the paper.

## DECLARATION OF INTERESTS

A.S.B. reports institutional grants outside of this work from AstraZeneca, Bayer, Biogen, BioMarin, Bioverativ, Novartis, Regeneron, and Sanofi.

## STAR★METHODS

Detailed methods are provided in the online version of this paper and include the following:

- KEY RESOURCES TABLE
- EXPERIMENTAL MODEL AND SUBJECT DETAILS
- METHOD DETAILS
  - Factor analysis of quantitative traits
  - Interpretation of latent factors
  - GWAS of latent factors using summary statistics
  - Multi-trait fine-mapping with flashfmZero
- QUANTIFICATION AND STATISTICAL ANALYSIS
  - GWAS and conditional analyses of blood cell traits and their latent factors
- ADDITIONAL RESOURCES

## SUPPLEMENTAL INFORMATION

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

# STAR★METHODS

## KEY RESOURCES TABLE

| REAGENT or RESOURCE | SOURCE | IDENTIFIER |
|---|---|---|
| **Deposited data** | | |
| INTERVAL summary statistics of blood cell traits | Akbari et al.[15] | http://ftp.sanger.ac.uk/pub/project/humgen/summary_statistics/sysmex_blood_cell_genetics |
| INTERVAL summary statistics of blood cell traits | Astle et al.[16] | https://www.phpc.cam.ac.uk/ceu/haematological-traits/ |
| INTERVAL summary statistics of NMR traits | Karjalainen et al.[18] | http://phpc.cam.ac.uk/ceu/lipids_metabolism/ |
| Latent factor GWAS summary statistics | This paper | https://www.ebi.ac.uk/gwas/; NHGRI-EBI GWAS Catalog: GCST90559243-GCST90559267 |
| **Software and algorithms** | | |
| R 4.3.1 | R Core Team 2023[40] | https://www.r-project.org/ |
| data.table | The R Foundation | https://cran.r-project.org/web/packages/data.table/index.html |
| dplyr | The R Foundation | https://cran.r-project.org/web/packages/dplyr/index.html |
| reshape2 | The R Foundation | https://cran.r-project.org/web/packages/reshape2/index.html |
| stringr | The R Foundation | https://cran.r-project.org/web/packages/stringr/index.html |
| tidyr | The R Foundation | https://cran.r-project.org/web/packages/tidyr/index.html |
| ggplot2 | The R Foundation | https://cran.r-project.org/web/packages/ggplot2/index.html |
| RColorBrewer | The R Foundation | https://cran.r-project.org/web/packages/RColorBrewer/index.html |
| BOLT-LMM v2.4.1 | Loh et al.[28] | https://data.broadinstitute.org/alkesgroup/BOLT-LMM/ |
| UKBB500K-Conditional-Analysis | Akbari et al.[15] | https://github.com/ParsaAkbari/UKBB500K-Conditional-Analysis |
| flashfmZero | This paper[39] | https://jennasimit.github.io/flashfmZero/jennasimit.github.io/flashfmZero/ |
| qctool v2 | code.enkre.net; Gavin Band | https://enkre.net/cgi-bin/code/qctool/dir?ci=trunk |
| forestplot | The R Foundation | https://cran.r-project.org/web/packages/forestplot/index.html |
| topr | The R Foundation | https://cran.r-project.org/web/packages/topr/index.html |
| networkD3 | The R Foundation | https://cran.r-project.org/web/packages/networkD3/index.html |
| psych (factor analysis package) | The R Foundation | https://cran.r-project.org/web/packages/psych/index.html |
| bigsnpR | The R Foundation | https://cran.r-project.org/web/packages/bigsnpr/index.html |
| mvsusieR | Zou et al.[13] | https://github.com/stephenslab/mvsusieR |
| **Other** | | |
| Variant Effect Predictor v113 (build 37) | McLaren et al.[41] | https://grch37.ensembl.org/info/docs/tools/vep/index.html |
| CMDKP | Common Metabolic Diseases Knowledge Portal | https://hugeamp.org/ |
| Custom code for the INTERVAL blood cell trait analyses that use complete data | This paper[38] | https://github.com/fz-cambridge/flashfmZero-INTERVAL-analysis |

*(Continued on next page)*

| *Continued* | | |
| --- | --- | --- |
| REAGENT or RESOURCE | SOURCE | IDENTIFIER |
| Custom code for the INTERVAL NMR analyses that use summary statistics | This paper[39] | https://jennasimit.github.io/flashfmZero/ |

## EXPERIMENTAL MODEL AND SUBJECT DETAILS

INTERVAL is a cohort of 48,725 generally healthy adult blood donors recruited through NHS Blood and Transplant, the English Blood Service, between 2012 and 2014.[17,42] The cohort was originally established for a clinical trial to assess the effect of variation in inter-donation time intervals on the health of blood donors.[43] The sample size of the cohort was determined in order to control the power to detect i) an operationally significant difference in donation rate and ii) a clinically significant difference in a measure of quality of life between trial arms.[17] The study was approved by the Cambridge East Research Ethics Committee and informed consent was obtained from all participants during recruitment.

Participants were genotyped by Affymetrix (Santa Clara, Ca, USA) with the UK Biobank Axiom array using DNA extracted from buffy coat by LGC Genomics (UK). Standard Affymetrix quality control (QC) procedures were applied to the resulting data, which excluded genotyping probes with low signal intensity, samples with low call rates and variants with low call rates or low confidence calls. Further QC procedures were applied—to the full dataset and within each genotyping batch—to remove rare variants, multiallelic variants, variants with a poor call rate, variants out of HWE and variants exhibiting allele frequency variability across batches. Variants were pruned to ensure no pair exhibited strong LD. Samples exhibiting evidence for contamination, excess heterozygosity, non-European ancestry or discordance of phenotypic and genotypic sex were removed. Subsequently, haplotype phases were imputed using SHAPEIT3 and missing genotypes were imputed from the 1000 Genomes Phase 3-UK10K reference panel using the PBWT.[44,45]

Extended complete blood counts (CBCs) were measured from EDTA treated blood samples taken from INTERVAL participants using two Sysmex XN haematology analysers at UK Biocentre (Stockport, UK). Because a flow cytometry channel of one instrument was misconfigured during the first 90 days of the study, data for some platelet variables are missing for some participants. The extended CBC produced by the Sysmex instrument measures various properties of the peripheral blood, including hemoglobin levels and properties of reticulocytes, mature red cells, platelets, neutrophils, eosinophils, basophils, monocytes and lymphocytes. These properties include cell concentrations, measures of cell maturity, properties of cell volume distributions and properties of the distributions of cell fluorescence and cell side-scatter measured by flow cytometry.

Each variable in the CBC was adjusted to remove variance explained by technical covariables including, the identity of the measuring instrument, the age of the blood sample at the time of measurement, the time of day of the measurement, time dependent instrumental drift and instrument recalibration events. Measurements taken more than 36 h after venipuncture were excluded. Subsequently, we adjusted each variable to remove variance explained by sex, menopausal status, age, smoking habits, drinking habits, log-height and log-weight. Finally measurements that were outliers in univariate and cell-type specific multivariate distributions were removed. The phenotypes were then rank inverse normalised.

More detailed descriptions of the QC procedures applied to the genotype and phenotype data are given by Akbari et al.[15] and Astle et al.[16]

A non-fasting serum sample was taken from INTERVAL blood donors before donation at the enrollment visit. High-throughput NMR spectroscopy-based metabolic profiling was used to quantify 230 metabolic traits from these serum samples. The metabolic profiles include routine lipids and individual lipids and their composition in 14 lipoprotein subclasses, fatty acids, amino acids, ketone bodies, glycolysis-related metabolites, and various other measures. Thirty-eight participants were removed from analysis due to a proportion of missing data >30% across lipid traits. Genetic analyses were implemented in BOLT-LMM and were adjusted for age, sex and 10 genetics PCs and were rank inverse normalised.

## METHOD DETAILS

### Factor analysis of quantitative traits

Let there be $P$ observed traits measured in $N$ individuals. Under the factor analysis model, we explain the variability in the $P$ traits by a smaller number of $K$ ($K < P$) latent (unobserved) factors that are related to the traits through a $P \times K$ factor loading matrix $\boldsymbol{L}$. Let $\mu_j$ be the mean of observed trait $j$ and $\boldsymbol{1}_N$ be an $N$-vector (column) of ones. Under this model, the observed $P \times N$ trait matrix $\boldsymbol{Y}$ is modeled by

$$Y = M + LF + \varepsilon$$

where $M$ is the $P \times N$ mean matrix ($M = (\mu_1, \ldots, \mu_P)^T \, 1_N^T$), $\boldsymbol{F}$ is the $K \times N$ matrix of factor scores and $\varepsilon$ is a $P \times N$ error term matrix with mean zero. A common approach to estimating the factor loadings matrix $\boldsymbol{L}$ is through maximum likelihood, which only requires the trait correlation matrix. When individual-level trait measurements are available, pairwise complete observations may be used to estimate the Pearson correlation coefficient between each pair of traits. Otherwise, GWAS summary statistics may be used in methods such as cross-trait LD score regression[27] to estimate the trait correlations.

We applied factor analysis in R[40] using the "fa" function in the psych library,[46] with the arguments fm = "ml", for a maximum likelihood factor analysis and rotate = "varimax". The varimax rotation preserves the orthogonality of latent factors (factor scores), so that they are uncorrelated.

We used Horn's parallel method, as implemented by the "fa.parallel" function in the psych library, to select the number of latent factors based on the observed trait data. In Horn's parallel method, eigenvalues are calculated from the observed data and from "noisy" random data. These two sets of eigenvalues are often compared in a scree plot, which displays the eigenvalue for each number of factors. The eigenvalues of the observed data will be larger than those from the random data until a certain point - this point where the observed data eigenvalues first become smaller than those from random data is the suggested number of factors.

Upon estimating the factor loadings, if individual-level trait measurements are available, the factor scores (latent factor values) may then be estimated by least squares as

$$\hat{F} = \left(L^T L\right)^{-1} L^T (Y - \underline{Y}),\qquad\text{(Equation 1)}$$

where $\underline{Y}$ is a $P \times N$ matrix of trait sample means, as an estimate of $M$. This is implemented in R using the "fa" function in the psych library.[46]

A trait correlation matrix is sufficient to construct latent factors by computing their loadings and to quantify the contribution of each factor (re-scaled factor loadings) to each observed trait. However, our initial objective is a first principles view, not only to compute the loadings, but also to compute the values of the latent factors for each individual (i.e., the factor scores). This requires individual-level data.

In order to calculate factor scores for each individual from the factor loadings and the measured traits, we only used participants that have measurements for all the measured traits. The application of imputation approaches such as Multivariate Imputation by Chained Equations (MICE)[47] was inappropriate, because the measurements were not missing independently by trait; subsets of individuals were missing certain platelet measurements, as described above in the INTERVAL cohort section. Consequently, rather than introducing noise through poor quality imputation, we opted to reduce the sample size. We later relax the requirement of complete data by deriving an estimate of latent factor GWAS effect estimates that only requires the factor loading matrix (obtained from factor analysis using the trait correlation matrix) and observed trait GWAS effect estimates. This avoids the need for individual-level data and allows flexibility to missing trait measurements.

### Factor analysis of blood cell traits using complete data

Initially, we used factor analysis to calculate latent factors from 99 blood cell traits from the INTERVAL cohort[17,42] using the observed trait measurements matrix for complete data, which enabled us to compute the values of the latent factors for each individual (i.e., the factor scores). Upon subsetting the INTERVAL study to participants who have measurements for each of the 99 blood cell traits, the final sample size was reduced from 43,059 to 18,310. Blood cell traits are categorised by broad cell type. Compound red blood cell, mature red blood cell, and immature red blood cell traits are all red blood cell traits. Compound white cell, lymphocyte, eosinophil, monocyte, basophil, and neutrophil traits are all white blood cell traits. A compound red cell trait is a trait that depends on measurements of mature red blood cells and reticulocytes, while a compound white cell trait is a trait that depends on measurements of lymphoid and myeloid white cells. A description of the blood cell traits, including their broad biological categories is given in Table S1.

Using the 18,310 x 99 matrix of blood cell trait measurements as input, we selected the number of latent factors by applying the "fa.parallel" function in the psych R package[46] with the argument fm = "ml" for a maximum likelihood factor analysis. The fa.parallel function implements Horn's method and outputs a scree plot that compares the eigenvalues calculated in the data and in random datasets. The number of latent factors is selected such that the data-calculated eigenvalues are larger than those based on the random datasets. This indicated that 25 latent factors was an optimal choice (Figure S1). The blood cell trait covariance matrix in the INTERVAL study amongst the complete samples is given in Table S4.

We applied factor analysis in R using the "fa" function in the psych library,[46] with the arguments nfactors = 25 for 25 latent factors, fm = "ml", for a maximum likelihood factor analysis and rotate = "varimax". The varimax rotation preserves the orthogonality of latent factors (factor scores), so that they are uncorrelated.

Custom code for factor analysis of the complete data is available at https://github.com/fz-cambridge/flashfmZero-INTERVAL-analysis.[38]

### Factor analysis of blood cell traits allowing for missing trait measurements

Relaxing the requirement of complete data for all individuals, we calculated the correlation matrix for the 43,059 INTERVAL study participants, where each pairwise Pearson correlation coefficient was calculated from the pairwise complete data for the trait pair. The number of observed measurements for each trait ranged from 29,084 to 40,466 with a median of 38,951 and the number of complete pairwise measurements ranged from 25,515 to 40,466 with a median of 36,338.

A scree plot (using "fa.parallel" in the psych package with the argument fm = "ml") indicated that 25 latent factors was an optimal choice (Figure S9), which is in agreement with our complete data analysis on the subset of 18,310 INTERVAL study participants (Figure S1). As the input data is a correlation (or covariance) matrix, rather than individual-level data, we must specify the number of pairwise complete observations, n.obs, which we set to the median, 36,338. However, we note that we found the latent factor results to be robust to the setting of n.obs, as identical results were obtained when setting n.obs to the maximum of 40,466.

We applied factor analysis in R[40] using the "fa" function in the psych library,[46] with the arguments nfactors = 25 for 25 latent factors, fm = "ml", for a maximum likelihood factor analysis and rotate = "varimax".

*Factor analysis of metabolic traits allowing for missing trait measurements*

Amongst 230 NMR metabolic panel traits measured in 40,849 INTERVAL study participants, we excluded one trait (ace; acetate) for a high proportion of missingness (not measured in 82% of the participants). These traits had high levels of correlation (9 pairs with correlation above 0.999 and 86 pairs with correlation above 0.99), resulting in a singular covariance matrix, which caused computational issues in factor analysis. Therefore, we sequentially removed the trait with the highest number of pairwise correlations above 0.99 until all correlations were less than or equal to 0.99; 45 traits were excluded from this processing. The proportion of missingness for the resulting 184 traits ranged from 0 to 39% with an upper quartile of 0.008. A description of the NMR metabolic traits, including their broad biological categories, is given in Table S11.

Using the trait covariance matrix (Table S14), we selected 21 latent factors based on Horn's parallel method and the "fa.parallel" function in the psych library of R (Figure S11). We applied factor analysis in R[40] using the "fa" function in the psych library,[46] with the arguments nfactors = 21 for 21 latent factors, fm = "ml", for a maximum likelihood factor analysis and rotate = "varimax".

Custom code for removal of traits with high missingness and high correlation, and for carrying out the factor analysis is available at https://jennasimit.github.io/flashfmZero/articles/Example_Part1.html.[39]

### Interpretation of latent factors

Let $L_{ij}$ be the factor loading of latent factor $j$ ($j=1, …,L$) for observed trait $i$ ($i=1,…P$). We define the contribution of latent factor $j$ to observed trait $i$ by $C_{ij} = \frac{L_{ij}^2}{\sum_{k=1}^{L} L_{ik}^2}$, to aid in mapping the contributions of the latent factors back to each observed trait. These scaled factor loadings indicate the proportion of variance in each observed trait $i$ that is explained by latent factor $j$, relative to the total variance explained jointly by the latent factors. That is, for each observed trait, the contributions from all factor loadings sum to one.

To understand which observed traits are explained by each latent factor, we collect observed traits that have the same top-contributing latent factor (Figure S4). We automate this in our "factor_contributions" function within the flashfmZero package,[39] which takes the factor loading matrix as input and returns the latent factor contributions (re-scaled factor loadings) and factor loading matrix with observed traits ordered by maximum contributing latent factor.

*Blood cell trait latent factor interpretations*

Concordance of the latent factors obtained from our 18k and 43k analyses was illustrated by plotting the latent factor contributions based on the 43k sample against those of the 18k sample (Figure S10). Latent factor contributions indicated that latent factors cluster blood cell traits grouped by broad cell-type into groups with common underlying variance generating mechanisms (Figure 1, source data in Table S2). We describe the principal effects of the latent factors on the blood cell traits in Table S3, in which we note the broad type of blood cell corresponding to the traits to which each latent factor makes major contributions and give descriptions of the effect of an increase in each latent factor on selected blood cell traits.

*Metabolic trait latent factor interpretations*

Latent factor contributions indicated that latent factors broadly corresponded to distinct biological categories but with varying specificity. Details of the scaled factor loadings that show the contributions for each latent factor to each NMR trait are given in Table S12. Interpretations of the latent factors relative to NMR traits are given in Table S13.

Custom code for calculating latent factor contributions using the "factor_contributions" function within the 'flashfmZero' package[39] is available within the flashfmZero package at https://jennasimit.github.io/flashfmZero/articles/Example_Part1.html. Analogous code was used for the blood cell traits.

### GWAS of latent factors using summary statistics

We remove the limitation of requiring complete data and derive an approach to calculating GWAS summary statistics for latent factors that only requires GWAS summary statistics of all observed traits, their covariance matrix, and the factor loading matrix. Briefly, the observed trait covariance (or correlation) matrix is used to obtain the factor loading matrix. Then, the factor loading matrix, observed trait covariance matrix, and observed trait GWAS summary statistics are used to compute the latent factor GWAS summary statistics (for each latent factor, variant effect estimates and their standard errors) directly. Our latent factor GWAS calculation is implemented in the "latentGWAS" function of the flashfmZero package.[39]

To take advantage of the lower dimension latent factors, we had conducted a GWAS on each latent factor using complete data. In a GWAS, we test trait $Y_j$ for marginal association with a genetic variant $x_k$, via the linear model

$$Y_{ij} = \alpha + \beta x_{ik} + \varepsilon_i,$$

where $Y_{ij}$ is the trait $j$ measurement at individual $i$ and $x_{ik}$ is the genotype of variant $k$ for individual $i$. In matrix form, for multiple traits with independent estimation of effect estimates (i.e., parallel GWAS of each trait and not a multi-trait GWAS), we have

$$Y^T = 1_N \alpha^T + x_g \beta^T + \varepsilon, \tag{Equation 2}$$

where $\alpha = (\alpha_1, ..., \alpha_P)^T$, $\beta = (\beta_1, ..., \beta_P)^T$, and $\varepsilon$ is a $N \times P$ error term matrix with mean zero. Likewise, when individual-level data are available to estimate factor scores we test latent factor $F_j$ for genetic association with variant $\boldsymbol{x_g}$ (vector of N genotype observations at $g^{th}$ variant) using

$$F_{ij} = \alpha_j^* + x_{ig}\beta_j^* + \varepsilon_{ij},$$

or in matrix form for multiple latent factors

$$F^T = 1_N \alpha^{*T} + x_g \beta^{*T} + \varepsilon, \qquad \text{(Equation 3)}$$

where $\alpha^* = (\alpha_1^*, ..., \alpha_K^*)^T$, $\beta^* = (\beta_1^*, ..., \beta_K^*)^T$, and $\varepsilon$ is a $N \times K$ error term matrix with mean zero.

Alternatively, if GWAS summary statistics are available for each trait, we may estimate the latent factor GWAS summary statistics, without the need for any individual-level data, since (1) and (3) give us

$$1_N \widehat{\alpha}^{*T} + x_g \, \widehat{\beta}^{*T} = (Y - \underline{Y})^T L (L^T L)^{-1}$$

Then, by substituting in estimates based on 2, we obtain

$$1_N \widehat{\alpha}^{*T} + x_g \, \widehat{\beta}^{*T} = (1_N \alpha^T + x_g \widehat{\beta}^T - \underline{Y}^T) L (L^T L)^{-1},$$

so that, upon matching coefficients, we have

$$\widehat{\beta}^* = (L^T L)^{-1} L^T \widehat{\beta}.$$

Therefore, upon estimating factor loadings $L$ via the trait correlation matrix, it is possible to estimate the latent factor GWAS effect estimates $\widehat{\beta}^*$ via the trait GWAS effect estimates $\widehat{\beta}$. Likewise, the covariance matrix of the latent factor effect estimates may be estimated by

$$Var(\widehat{\beta}^*) = (L^T L)^{-1} L^T Var(\widehat{\beta}) L (L^T L)^{-1},$$

where $[Var(\widehat{\beta})]_{ij} = Cov(Y_i, Y_j)\sqrt{Var(\widehat{\beta_i})Var(\widehat{\beta_j})}$, and $\sqrt{Var(\widehat{\beta_i})}$ is the standard error of the variant's effect estimate for observed trait $i$, as provided by the trait's GWAS summary statistics.

It follows that the latent factor genetic associations may be assessed via the Z-statistic,

$$Z_j = \frac{\widehat{\beta}_j^*}{\sqrt{\left[Var\left(\widehat{\beta}_j^*\right)\right]_{ij}}}, \text{for latent factor } j.$$

We note that this formulation removes the limitation of requiring complete data to estimate latent factor GWAS summary statistics.

### GWAS of latent factors of blood cell traits and of metabolic traits

Within R,[40] we applied the "latentGWAS" function of the flashfmZero package[39] to the observed trait GWAS summary statistics of the 99 blood cell traits[15,16] and of the 184 metabolic traits[18] in the same manner. For simplicity, we describe these steps for the metabolic traits and custom code is available within the flashfmZero package at https://jennasimit.github.io/flashfmZero/articles/Example_Part2.html.[39]

When calculating the latent factor GWAS summary statistics from the observed trait GWAS summary statistics, each variant must have the same effect allele across all traits. To simplify this harmonisation process, the function "harmoniseGWAS" is available in the flashfmZero package. This function also includes filtering of variants - we set the minimum MAF, minMAF = 0.005, and minimum INFO score, minINFO = 0.4. After harmonising the observed trait GWAS, the latent factor GWAS were then calculated using the "latentGWAS" function in the flashfmZero package, which outputs a list of GWAS summary statistics for each latent factor.

### Multi-trait fine-mapping with flashfmZero

The multi-trait fine-mapping method, flashfm,[4] leverages information between traits while allowing for multiple causal variants that are not necessarily shared between traits. It is flexible to missing trait measurements. When there are shared causal variants, flashfm has been shown to improve fine-mapping resolution and increase the number of high-confidence variants, compared to single-trait fine-mapping.[4,48] Otherwise, it gives comparable results to single-trait fine-mapping.

Flashfm requires the trait correlation matrix and is currently limited to six traits at most. However, we take advantage of the uncorrelated latent factors that result from using a varimax rotation, resulting in a diagonal correlation matrix. Under this condition, we have extended flashfm to multiple-trait fine-mapping of an unlimited number of (uncorrelated) latent factors. We call this extended method flashfmZero and implement it within R[40] in the flashfmZero function of the flashfmZero package,[39] which can be combined with any single-trait fine-mapping method that outputs multi-SNP model posterior probabilities, such as JAM[21] FINEMAP,[23] and FiniMOM.[49] Within the flashfmZero package, we also provide the wrapper function FLASHFMZEROwithJAMd that runs our dynamic version of

JAM (dynamically select maximum number of causal variants based on the data) together with flashfmZero; this is the function that we used in all analyses.

For $M$ uncorrelated traits, the joint Bayes' factor $BF^M$ can be expressed as the product of the marginal trait $BF_j$, $j=1, \ldots M$. Without loss of generality, the next steps focus on $M=2$ traits. As in flashfm,[4] the joint prior probability $p_{i,j}^{(1,2)}$ for models $M_i^{(1)}$ and $M_j^{(2)}$ for traits 1 and 2, respectively, is defined as the product of the marginal prior probabilities when there is no model overlap of causal variants and is upweighted when there is sharing. That is, $p_i p_j \kappa^{1\{M_i^{(1)} \cap M_j^{(2)} \neq \varnothing\}} \tau_{i,j}$, where $\kappa$ is a sharing parameter and $\tau_{ij}$ is a correction factor that guarantees that the prior probability of traits having particular model sizes is consistent for different values of $\kappa$; both parameters are derived in a combinatorial manner as in flashfm.[4] It follows that the trait-adjusted posterior probability for model $M_i^{(1)}$ of trait 1 is calculated from

$$Pr\left(M_i^{(1)}|Data\right) \propto PP_i \sum_j PP_j \kappa^{1\left\{M_i^{(1)} \cap M_j^{(2)} \neq \varnothing\right\}} \tau_{i,j}$$

which is easily generalised to any number of traits $M$ due to the traits being uncorrelated.

### Fine-mapping of associations of blood cell traits and their latent factors

Within the complete INTERVAL data subset for blood cell traits, we first investigated gains from fine-mapping association signals using latent factors that are uncorrelated by construction, over fine-mapping association signals using a larger number of correlated traits. We constructed regions based on the latent factor association signals. For each latent factor, we used distance-based clumping to identify lead SNPs with a distance of at least 250kb, which were then centered ±250kb to form regions. Regions that overlapped amongst traits were merged. We further expanded our regions by integrating them with those from fine-mapping signals from 29 blood cell traits in UK Biobank[6] and merging any that overlapped, so that our regions contained those used in the UK Biobank fine-mapping. This resulted in 217 regions with lengths ranging from 500,000bp to 2,996,725bp.

Within these regions, we fine-mapped genome-wide association signals ($P < 5 \times 10^{-8}$, MAF>0.005) with all the latent factors and with all the blood cell traits that have a contribution of at least 20% from these latent factors (Figure S4). Single-trait fine-mapping of latent factors and blood cell traits was carried out with JAMdynamic,[22] which is an extension of JAM[21] that dynamically selects the maximum number of causal variants based on the data. When multiple latent factors had a signal in a region, we also used our zero-correlation version of flashfm, as available in the wrapper function FLASHFMZEROwithJAMd(https://jennasimit.github.io/flashfmZero/).

For fine-mapping, we used an LD matrix calculated from the 18,310 participants in the INTERVAL cohort that contributed to the GWAS. In particular, within R[40] we used the bigsnpR library to read in bgen genotype files that were previously subset to the required region by using qctool (https://enkre.net/cgi-bin/code/qctool/dir?ci=trunk). We then used the alignGWAS function within the flashfmZero package to ensure that variants in all GWAS are aligned to the same alleles as in the genotypes file for LD calculation. We used best-guess genotypes with a certainty threshold of 0.2, such that the genotype at a variant took on values 0,1, or 2 if their dosage was within 0.2 of the respective value; otherwise, the genotype was coded as NA in the correlation calculation; this process for is available in the LDqc function of the flashfmZero package. For each variant, we calculated the proportion of individuals with non-missing best-guess genotypes, and excluded any variants that had a non-missing proportion below 80%. Finally, we calculated LD using the bigcor function of bigsnpR to calculate pairwise-complete correlations.

In our comparisons of the fine-mapping resolution of the three approaches: (i) 'JAM blood cell trait' (JAMdynamic on each blood cell trait); (ii) 'JAM latent factor' (JAMdynamic on each latent factor); (iii) 'flashfm latent factor' (flashfmZero on each set of latent factors), we considered the CS99 size and variants with PP > 0.90, matching on traits (Table S8). That is, when comparing blood cell trait results to latent factor results, we match each blood cell trait to the latent factor that is the highest contributor to it. Custom code is available at https://github.com/fz-cambridge/flashfmZero-INTERVAL-analysis.[38] All genetic physical positions are given in GRCh37 coordinates.

We considered a variant to be a high-confidence causal variant if it had MPP > 0.90 and cross-checked our results with the high-confidence causal variants (MPP > 0.95) from the UK Biobank analysis[6] for validation; as our sample size is substantially smaller than that of UK Biobank, we used a slightly lower threshold when defining high-confidence. We identified 53 regions where a high-confidence variant was detected by either single-trait or multi-trait fine-mapping of the latent factor association signals and also detected in UK Biobank. Amongst these 53 regions, 17 regions are not comparable because our analyses included only latent factors that are linked to extended Sysmex traits, whereas the UK Biobank analyses did not include all the extended Sysmex traits. Therefore, we focused on 36 regions in cross-checking our latent factor fine-mapping results with those of the UK Biobank study. We also note that there were 9 regions where no high-confidence variants were identified by our latent factor analyses, but there was prioritisation by 'JAM blood cell trait'—in 5 of these regions there was alignment with the UK Biobank results and in the remaining 4 regions there was not an exact match in the high-confidence variants selected in INTERVAL and UK Biobank (Table S8). We highlight fine-mapping in the regions harboring *SMIM1* (Figure 6), *PIEZO1* (Figures S5 and S6), and *TMCC2* (Figures S7 and S8).

For comparison purposes, we applied mvSuSiE,[13] using the "mvsusie_rss" function in mvsusieR, to two regions where the likely causal variant has biological support. In our implementation of mvSuSiE we used the canonical prior and followed the author's suggestion of estimating the residual variance using the variants with absolute $Z$ score below 2 for all traits - this required using the mvsusieR functions "cov_canonical" and "create_mixture_prior". Within mvsusie_rss we also set coverage to 0.99. Details of these comparison results are in Table S9.

We selected eleven regions in which we had previously fine-mapped genetic association signals with blood cell traits and their latent factors using the complete data (18k) and repeated the fine-mapping using the summary statistics approach in the larger sample (43k) (Table S10). To calculate latent factor GWAS we first harmonised all observed trait GWAS so that all variants are aligned to the same allele across GWAS (using the "harmoniseGWAS" function in flashfmZero), then applied the "latentGWAS" function within flashfmZero. As for our previous analysis, we also used harmoniseGWAS to ensure that variants in all GWAS were aligned with the same alleles in the genotype file used for LD calculation. We included all variants having MAF >0.005 and INFO >0.4 in each GWAS, and the latent GWAS calculation only includes variants that are present for all observed trait GWAS - this was not an issue as all traits were measured in the same cohort.

To aid in summarising the fine-mapping results from each trait and latent factor across all regions, we provide the "FMsummary_table_general" function in the flashfmZero package. An example pipeline is available at https://jennasimit.github.io/flashfmZero/articles/Example_Part2.html.[39]

### Fine-mapping of associations of metabolic traits and their latent factors

We fine-mapped associations in eleven 1MB-regions in which both latent factors and NMR traits had genome-wide significant (GWS) association signals. As in the blood cell trait fine-mapping, we compared the fine-mapping resolution of the three approaches: (i) 'JAM metabolic trait' (JAMdynamic on each metabolic trait); (ii) 'JAM latent factor' (JAMdynamic on each latent factor); (iii) 'flashfm latent factor' (flashfmZero on each set of latent factors), we considered the CS99 size and variants with PP > 0.90, matching on traits.

All details for LD and latent GWAS calculations follow the same steps described above for blood cell traits.

Custom code for our fine-mapping analyses is available within the flashfmZero package at https://jennasimit.github.io/flashfmZero/articles/Example_Part2.html.[39]

## QUANTIFICATION AND STATISTICAL ANALYSIS

### GWAS and conditional analyses of blood cell traits and their latent factors

Within the 18,310 individuals complete data subset from the INTERVAL cohort, we first used an inverse normal rank transformation on each of the 99 blood cell traits using the "RankNorm" function of the RNOmni R library. We then calculated latent factor scores for each of our 25 latent factors by applying factor analysis (using the fa function in the R psych library) to the individual-level transformed trait data (only individuals with a measurement for each of the 99 traits were included). By providing individual-level data instead of a covariance or correlation matrix, the latent factor scores are output with the factor analysis results within the scores component of the output list. These latent factor scores were also transformed using an inverse normal rank transformation on each of the 25 latent factors.

Each of the 25 latent factors and 99 blood cell traits was tested for genetic associations within the sample of 18,310 individuals from the INTERVAL cohort using BOLT-LMM[35] with the following covariates: dummy variables indicating the donor clinic at which the blood sample was taken and the score vectors corresponding to the leading ten principal components of genetic variation in the study sample. This follows the approach taken in a previous large-scale GWAS of blood cell traits (that included the INTERVAL cohort) in 173,480 European descent individuals[16] and a GWAS of flow cytometry derived (Sysmex) blood cell traits in 41,515 INTERVAL cohort participants[15] (https://github.com/ParsaAkbari/UKBB500K-Conditional-Analysis).

All the traits were inverse normal-rank transformed prior to running BOLT-LMM. We report the infinitesimal mixed model association test $p$-value ("P_BOLT_LMM_INF") of each genetic variant with each trait and details of lead variants from latent factor GWAS, including levels of association for blood cell traits that have contributions from each latent factor are available in Table S5. All genetic physical positions are given in GRCh37 coordinates. The GWAS summary statistics for our 25 latent factors from this complete data subset are available at the NHGRI-EBI GWAS Catalog (https://www.ebi.ac.uk/gwas/) under accession numbers GCST90559243-GCST90559267.

To identify potentially novel association signals in our latent factor GWAS of 18,310 individuals, we conditioned on all the lead variants identified in the previously published large-scale GWAS of common blood cell traits[16] and the GWAS of Sysmex blood cell traits.[15]

We obtained a list of unique variants that are genome-wide significant for any of the 99 blood cell traits, through LD clumping ($r^2$ > 0.6) on the merged list of associated variants. Then, at each unique variant we recorded the number of blood cell traits that were associated with the variant; if the variant was not associated with a blood cell trait, but it had a tag variant (in the same clump) that was associated, the trait was enumerated. To identify unique variants missed by blood cell traits, we enumerated the unique independent genome-wide significant variants obtained only by latent factors, based on LD clumping ($r^2$ > 0.6) of their associated variants, allowing for the variant or one of its tag variants to be associated. Counts of unique clumps with genome-wide significant variants identified by blood cell traits and/or latent factors are given in Table S6.

Details of conditional lead variants associated with latent factors of blood cell traits are given in Table S7 and highlighted in Figure 4 (with additional details in Figure S2) and Figure S3. Within Table S7 we include the most serious consequence of each variant, as annotated by the Variant Effect Predictor (VEP)[41] and list association evidence for these variants from previous blood cell trait publications, as available in the Common Metabolic Diseases Knowledge Portal (https://hugeamp.org/).

## ADDITIONAL RESOURCES

This work involves data collected from the INTERVAL BioResource, which involves participants from the INTERVAL trial (ISRCTN 24760606).

