## [Document S2. Transparent peer review records for Zhou et al · Cell Genomics]

Improved genetic discovery and fine-mapping resolution through multivariate latent factor analysis of high-dimensional traits

Author list

Feng Zhou, William J Astle, Adam S Butterworth, Jennifer L Asimit

Summary**Initial submission:** Received : August 23rd 2024

Scientific editor: Judith Nicholson

First round of review: Number of reviewers: 2
Revision invited : September 23rd 2024
Revision received : December 23rd 2024**Second round of review:** Number of reviewers: 2
Accepted : March 14th 2025**Data freely available:** Yes**Code freely available:** Yes

This transparent peer review record is not systematically proofread, type-set, or edited. Special characters, formatting, and equations may fail to render properly. Standard procedural text within the editor's letters has been deleted for the sake of brevity, but all official correspondence specific to the manuscript has been preserved.

Referees' reports, first round of review

Reviewer 1

The study presents flashfmZero, a zero-correlation multi-trait fine-mapping approach/tool, capable of working with any number of latent factors derived from factor analysis of high-dimensional traits. In this study, the authors applied the method to blood cell traits in the INTERVAL cohort, demonstrating the potential power of latent trait fine-mapping for genetic discovery.

Major Comments:

1. Introduction

- The introduction lacks depth in describing the complexity of blood cell traits as a model. The authors should elaborate on the biological diversity of blood cell traits, previous GWAS efforts, and known risk loci to frame the significance of selecting this specific model. Including a detailed justification for using blood traits would strengthen the context and relevance of the study.
- The description of current multi-trait fine-mapping methods is somewhat limited, focusing primarily on flashfm. The introduction should also summarize and compare other relevant approaches (e.g., mvSuSiE, CAFEH) to show how flashfmZero differentiates itself and potentially outperforms these established methods. This would provide a more comprehensive background and solidify the motivation for developing flashfmZero.

2. Methodology and application

- The authors mention 25 latent factors derived from 99 blood cell traits, but the rationale for selecting these latent factors is unclear. A more detailed explanation of how the factors were chosen and what specific biological traits they represent would improve the clarity of the study.
- It would be beneficial to include a flowchart that illustrates the entire process—from input to output—along with an overview plot depicting key comparisons (e.g., latent factors vs. raw blood traits and single-trait vs. multi-trait fine-mapping). Visual aids would greatly enhance the readers' ability to follow the methodology and understand the experimental design.

3. Results

- This section contains inconsistencies that should be addressed. Specifically, the

statement "we identified five distinct genome-wide significant..." conflicts with the following statements that refer to four variants. A clearer explanation, perhaps through restructuring the results presentation or cross-referencing the findings with Table 1, would resolve this discrepancy.

- While the authors mention genetic associations with specific variants (e.g., IL5RA and rs9310935), a more thorough description of these associations and the potential functional relevance of these genes would strengthen the biological interpretation of the results. Furthermore, details on the nearby genes and pathways related to the associated variants should be elaborated to enhance the credibility and scientific significance of these findings.

4. Performance evaluation

- While flashfmZero demonstrates strong performance with blood cell traits, validation on other high-dimensional multi-trait datasets is necessary to confirm its generalizability. The inclusion of additional datasets—potentially incorporating imputed traits—would not only strengthen the analysis but also offer insights into the tool's versatility across different contexts.

- The study only retains a portion of the dataset for efficiency reasons, raising questions about the required data volume. A systematic benchmarking analysis should be conducted to identify the minimum sample size necessary for maintaining fine-mapping accuracy. This would provide valuable guidance for future users of flashfmZero on the minimum dataset thresholds.

- To demonstrate the superiority of flashfmZero, the authors should compare its performance with alternative methods, such as mvSuSiE (an extension of SuSiE, a preprint, PMID37425935), CAFEH (PMID35085493), or other multi-trait fine-mapping approaches. Using both real and simulated datasets would give a more comprehensive evaluation of flashfmZero's advantages and limitations.

Minor Comments:

1. Terminology and abbreviations

In the results section, blood trait abbreviations (e.g., RBC-SFL-DW) should be directly defined in the text, rather than referring readers to supplementary tables. This would enhance accessibility and prevent unnecessary interruptions for readers unfamiliar with the acronyms.

2. Statistical descriptions

The results section should include more interpretation of the descriptive statistics presented. For example, statements like "76% of the comparisons" and "strictly smaller in 58% of comparisons" need more explanation to clarify what constitutes a "strict" comparison and how these percentages are defined. Including thresholds or clarifying metrics will make the statistical claims easier to understand.

3. Improving figures and visual aids

- Figure 1: The abbreviations used (e.g., IG%) should be clarified in the figure legend to ensure that readers without specific knowledge of blood cell traits can understand the terms used.

- Figure 4: The axes need clearer labeling. Providing detailed explanations for the x-axis and y-axis would improve readers' ability to quickly interpret the data presented in the plot.

- Figure 6: The distinction between SNP groups should be explained more thoroughly. The use of letters (A-C) versus numbers (0) should be justified or clarified to help readers understand the coding system applied to these SNP groups.

4. Methods

The methods section mentions selecting the optimal number of latent factors (i.e., 25), but lacks a detailed description of how this was determined.

Supplementary figures illustrating the process for choosing the optimal number of latent factors would make the methodology more transparent.

5. R Packages and code accessibility

The flashfmZero package has installation issues on MacOS, specifically relating to the "gfortran" library. The authors should provide a troubleshooting guide for users who encounter this problem, along with any relevant resources to ensure smooth installation across operating systems. Additionally, some users have reported errors when running showcase examples on the flashfmZero website (e.g., "maxcv_stop" errors). Ensuring the reproducibility of these examples and providing clear input data formats on GitHub would increase the accessibility and usability of the method for a broader audience.

6. Discussion and conclusion

The current discussion does not explore the potential for applying flashfmZero to other categories of high-dimensional traits or disease models. Including a discussion of the tool's applicability to other biological domains, such as rare diseases, or its utility in drug target prioritization (such as Open Targets Genetics / PMID34711957 and Priority index / PMID31253980) would broaden the impact of the research and open avenues for future exploration.

Reviewer 2

The authors present flashfmZero, a novel methodology for fine-mapping in genome-wide association studies (GWAS) of high-dimensional traits. This approach builds upon the existing flashfm method and offers significant advantages in elucidating GWAS findings, particularly in the context of correlated traits and multi-morbidities. By leveraging latent factors derived from multiple related traits, flashfmZero aims to enhance both discovery power and fine-mapping precision.

1. The method's use of latent factors to capture shared biological mechanisms across multiple traits is innovative and potentially powerful.
2. The results suggest that flashfmZero can produce more precise credible sets compared to single-trait analyses, which could lead to more efficient identification of causal variants.
3. The ability to detect signals that are not genome wide significant with several traits and are minimally associated demonstrates the method's potential to uncover previously undetected genetic associations.

Here are some points for concerns/clarification:

1. Since the method can only be applied in the individual level datasets, it presents with a big challenge in the way GWAS is conducted in modern times, i.e. meta-analyzing data from multiple resources and then finamapping. Authors should present with some future directions and possibilities on how this method can be made applicable to non-individual level datasets. Potential avenues could include developing summary-statistic based approximations or potentially exploring federated learning approaches to maintain privacy while leveraging multi-cohort data.
2. The paper lacks discussion on how flashfmZero performs across different

genetic ancestry groups. Authors should address how the method accounts for population structure and whether it can be applied to or interpreted for multi-ancestry studies.

3. Comparisons with ancestry-specific fine-mapping methods like MR-MEGA or extensions of MeSuSie for multi-trait analysis could provide valuable insights.

4. Exploring the use of ancestry-specific LD panels within the flashfmZero framework could enhance its applicability to diverse populations.

5. The method is only applied on quantitative traits, is it applicable for binary traits. Please elaborate.

6. Its hard to understand how latent factors can be constructed for a new dataset. I looked through the code in Github and it only shows the application from this paper. To make this tool more broadly available, it is important to expand the Github to explain its applicability to broader questions.

7. Interpretation of latent factor result si also difficult, Authors should ffer guidelines on interpreting results and selecting appropriate parameters.

Authors' response to the first round of review

We appreciate the constructive comments from the reviewers and feel that our revised version is an improvement over our original manuscript, thanks to their comments. All comments are addressed individually, and we list some of the significant changes here.

We have derived an approach for latent factor genome-wide association study (GWAS) computation that requires only summary-level data. This removes the need for data where all individuals have measurements for all traits, and widens the use of our approach to summary-level datasets freely available on-line. Using summary-level blood cell trait data from INTERVAL, allowing for missing trait measurement, we apply our new approach and show agreement with our previous analysis on complete data.

To show broad applicability of our approach to other traits, we analyse nearly 200 NMR metabolic traits from the INTERVAL study as a second applied example. We provide interpretations of the latent factors and illustrate the gains of our joint latent factor fine-mapping approach in the application of flashfmZero to several regions, including dissecting multiple signals from multiple traits in the same region.

We provide additional functions in our flashfmZero R package (<https://jennasimit.github.io/flashfmZero/>) that help users with preparing data for analysis, the interpretation of latent factors, estimate latent factor GWAS summary statistics, and construct table summaries of fine-mapping results from the many traits and latent factors. Fully annotated scripts that we used in our analyses are also provided with guidance for users to adjust to their own data; these are available as articles from the flashfmZero web page listed previously.

Reviewers' Comments:

Reviewer #1: The study presents flashfmZero, a zero-correlation multi-trait fine-mapping approach/tool, capable of working with any number of latent factors derived from factor analysis of high-dimensional traits. In this study, the authors applied the method to blood cell traits in the INTERVAL cohort, demonstrating the potential power of latent trait fine-mapping for genetic discovery.

Major Comments:**1. Introduction**

- The introduction lacks depth in describing the complexity of blood cell traits as a model. The authors should elaborate on the biological diversity of blood cell traits, previous GWAS efforts, and known risk loci to frame the significance of selecting this specific model. Including a detailed justification for using blood traits would strengthen the context and relevance of the study.

While we are keen to ensure that this is a paper that focuses on our multi-trait fine-mapping tool, using blood cell traits (and now also metabolic traits) as exemplar datasets only, we have nevertheless expanded our Introduction to include more detail on the GWAS of blood cell traits:

"A complete blood count (CBC) report is an example of a multivariate phenotype in which correlation between the component traits arises, in part, because of a common dependence on variation in one or more biological processes. All types of blood cells derive from a common stem cell type, the haematopoietic stem cell (HSC) and different types of blood cells interact, for instance in haemostasis and in immune responses. CBCs include measurements of haemoglobin concentrations and measurements of the blood concentrations of reticulocytes, mature red blood cells, platelets and the different types of white blood cells. Additionally, they often contain

measurements of the mean cell volumes of several cell types. GWAS of CBC traits have been conducted using samples of hundreds of thousands of participants, identifying hundreds of associations with common and rare genetic variants. Many of these associations are shared by biologically related traits. For example, genetic variants that increase mean platelet volume, usually also reduce platelet count, presumably because the proportion of blood volume occupied by platelets is physiologically regulated(Collins et al. 2021). The missense variant rs3184504 in SH2B3, which encodes lymphocyte adapter protein (LNK), is associated with traits measuring properties of reticulocytes, mature red cells, neutrophils, eosinophils, basophils, lymphocytes, and monocytes(Vuckovic et al. 2020). LNK encodes an adaptor protein that regulates cytokine signaling in HSCs and plays a crucial role in HSC self-renewal and the differentiation of all the major blood cell lineages(Quantification of Self-Renewal Capaci...; Seita et al. 2007). Because CBCs typically contain at least two dozen traits measured simultaneously, many of which are genetically and biologically correlated, they provide an ideal testing ground for multi-trait association methods. ”

- The description of current multi-trait fine-mapping methods is somewhat limited, focusing primarily on flashfm. The introduction should also summarize and compare other relevant approaches (e.g., mvSuSiE, CAFEH) to show how flashfmZero differentiates itself and potentially outperforms these established methods. This would provide a more comprehensive background and solidify the motivation for developing flashfmZero.

Thank you for this helpful comment and we agree that additional background provides stronger motivation. We have added the following within the Introduction:

“Current multi-trait fine-mapping methods that allow for multiple causal variants are not scalable to high-dimensional traits. CAFEH(Arvanitis et al. 2022) and mvSuSiE(Zou et al. 2024) are multi-trait fine-mapping extensions of SuSiE(Wang et al. 2020) fine-mapping: CAFEH assumes the traits are independent, while mvSuSiE models trait correlations. In addition, CAFEH can account for missing trait measurements, whereas mvSuSiE requires complete data.. Flashfm[4] multi-trait fine-mapping accounts for between trait correlations and leverages information between traits in a Bayesian framework, allowing for missing trait measurements; the prior on the model space allows traits to have shared and distinct causal variants and upweights multi-trait models with shared causal variants. Consequently, flashfm has improved resolution when traits share causal variant(s), but otherwise gives similar results to single-trait fine-mapping[4]. CAFEH and flashfm provide trait-specific posterior probabilities of causality for each trait, in the same way as MTAG multi-trait GWAS(Turley et al. 2018). In contrast mvSuSiE outputs a posterior probability that each variant is causally associated with at least one trait and relies on a second metric (local false sign rate) to infer which the associated traits are.”

2. Methodology and application

- The authors mention 25 latent factors derived from 99 blood cell traits, but the rationale for selecting these latent factors is unclear. A more detailed explanation of how the factors were chosen and what specific biological traits they represent would improve the clarity of the study.

Thank you for this clarifying comment. We have re-structured the Results section so that we reference Table S3 after explaining the selection of 25 latent factors and also include a scree plot (Figure S1). Table S3 maps latent factors to broad blood cell trait categories and gives descriptions on the effect of each latent factor on particular blood cell traits. We also provide more details in Methods. Finally, we have included a new function (factor_contributions) in our flashfmZero package, which calculates the latent factor contributions from the factor loading matrix and outputs the matrices in an easier-to-interpret form, grouping observed traits by highest contributing latent factor.

In particular we have added the following to Results:

“Using Horn’s parallel method, we decided to use a model including 25 statistically uncorrelated latent factors. (Figure S1, Methods).”

and

“We describe the principal effects of the latent factors on the blood cell traits in Table S3, in which we note the broad type of blood cell corresponding to the traits to which each latent factor makes major contributions and give descriptions of the effect of an increase in each latent factor on selected blood cell traits.”

and the following to Methods:

“Let there be P observed traits measured in N individuals. Under the factor analysis model, we explain the variability in the P traits by a smaller number of K ($K < P$) latent (unobserved) factors that are related to the traits through a $P \times K$ factor loading matrix L . Let μ_j be the mean of observed trait j and $\mathbf{1}_N$ be a N -vector (column) of ones. Under this model, the observed $P \times N$ trait matrix Y is modelled by

$$Y = M + LF + \varepsilon$$

where M is the $P \times N$ mean matrix ($M = (\mu_1, \dots, \mu_P)^T \mathbf{1}_N^T$), F is the $K \times N$ matrix of factor scores and ε is a $P \times N$ error term matrix with mean zero. A common approach to estimating the factor loadings matrix L is through maximum likelihood, which only requires the trait correlation matrix. When individual-level trait measurements are available, pairwise complete observations may be used to estimate correlation between each pair of traits. Otherwise, GWAS summary statistics may be used in methods such as cross-trait LD score regression[30] to estimate the trait correlations.

The number of latent factors is selected based on the observed trait data, and this decision is often made based on a scree plot, which relies on eigenvalues. A scree plot displays the eigenvalue for each factor, in decreasing order of magnitude. As eigenvalues measure the amount of variance accounted for by each factor, the point in the plot before the plateau starts indicates the smallest factor that contributes a meaningful amount of variance.

Upon estimating the factor loadings, if individual-level trait measurements are available, the factor scores (latent factor values) may then be estimated by least squares as

$$\hat{F} = (L^T L)^{-1} L^T (Y - \underline{Y}), \quad (1)$$

where \underline{Y} is a $P \times N$ matrix of trait sample means, as an estimate of M .

”

and a new section within Method Details:

“Interpretation of latent factors

Let L_{ij} be the factor loading of latent factor j ($j=1,\dots,L$) for raw trait i ($i=1,\dots,P$). We define the contribution of latent factor j to raw trait i by $C_{ij} = \frac{L_{ij}^2}{\sum_{k=1}^L L_{ik}^2}$, to aid in mapping the contributions of the latent factors back to each raw trait. These scaled factor loadings indicate the proportion of variance in each observed trait i that is explained by latent factor j , relative to the total variance explained jointly by the latent factors. That is, for each observed trait, the contributions from all factor loadings sum to one.

To understand which observed traits are explained by each latent factor, we collect observed traits that have the same top-contributing latent factor. We automate this in our “factor_contributions” function within the flashfmZero package[34], which takes the factor loading matrix as input and returns the latent factor contributions (re-scaled factor loadings) and factor loading matrix with observed traits ordered by maximum contributing latent factor.

“

- It would be beneficial to include a flowchart that illustrates the entire process—from input to output—along with an overview plot depicting key comparisons (e.g., latent factors vs. raw blood traits and single-trait vs. multi-trait fine-mapping). Visual aids would greatly enhance the readers' ability to follow the methodology and understand the experimental design.

We agree that this would be helpful and have included two new figures to address this.

Figure 2 consists of two flow charts that illustrate the estimation of latent factor GWAS summary statistics using (A) individual-level data; (B) GWAS summary statistics from observed traits and their covariance matrix.

Figure S4 illustrates (A) interpretations of latent factors based on contributions to observed traits; (B) flow diagram of our fine-mapping process depicting the key comparisons.

3. Results

- This section contains inconsistencies that should be addressed. Specifically, the statement "we identified five distinct genome-wide significant..." conflicts with the following statements that refer to four variants. A clearer explanation, perhaps through restructuring the results presentation or cross-referencing the findings with Table 1, would resolve this discrepancy.

We thank the reviewer for this comment and we have re-structured to resolve this - in addition to rs9310935, there were five additional distinct genome-wide significant associations and amongst these six variants there were four common and two low-frequency variants:

"Despite analysing a relatively small sample (18,310 participants) compared to previously published GWAS of blood cell traits, we identified six distinct genome-wide significant associations with the latent factors, after conditioning on the 3,559 lead variants from recent large GWAS of complete blood cell traits[11] and Sysmex extended blood count traits[10]. These included the association between ML8 and rs9310935 in IL5RA. None of the six associated variants showed genome-wide significant evidence for association in our univariate analyses of the blood cell traits in the same participants (Methods). In summary, amongst the six variants exhibiting novel associations, four of the variants were common (MAF>0.01) with different likely causal genes and two were low-frequency (0.003 < MAF < 0.01) from the same likely causal gene (Table 1, Table S7)."

We have carefully read through the Results and have also added reference to Table S4 in our discussion of rs6064377.

- While the authors mention genetic associations with specific variants (e.g., IL5RA and rs9310935), a more thorough description of these associations and the potential functional relevance of these genes would strengthen the biological interpretation of the results. Furthermore, details on the nearby genes and pathways related to the associated variants should be elaborated to enhance the credibility and scientific significance of these findings.

We have now included two columns in Table 1 to list the likely causal gene and biological support for the likely causal gene.

In the text our illustration with IL5RA is expanded by adding

"IL5RA encodes the interleukin-5 receptor alpha subunit of a heterodimeric cytokine receptor found on the surface of eosinophils and basophils. Interleukin-5 signalling induces the differentiation and maturation of eosinophils in the bone marrow. Therapies specifically targeting this protein, such as benralizumab, are effective at blocking interleukin-5 signalling, reducing basophil and eosinophil counts through apoptosis, and therefore treating eosinophilic airway diseases such as severe eosinophilic asthma."

and for FAM210B we have added

"rs6064377 lies near the gene *FAM210B*, which encodes the protein Family With Sequence Similarity 210 Member B, a mitochondrial membrane protein which is activated by GATA-1, a critical transcription factor for erythroid differentiation. *FAM210B* is thought to play a key role in regulating mitochondrial iron import to allow heme synthesis, thereby regulating erythropoiesis and iron transport, consistent with the associations seen with red blood cell traits[15]."

4. Performance evaluation

- While flashfmZero demonstrates strong performance with blood cell traits, validation on other high-dimensional multi-trait datasets is necessary to confirm its generalizability. The inclusion of additional datasets—potentially incorporating imputed traits—would not only strengthen the analysis but also offer insights into the tool's versatility across different contexts.

We have now developed an approach that uses summary-level data to estimate latent factor GWAS, removing the need for complete data. We have applied this approach to blood cell traits from the INTERVAL study, allowing for missing trait measurements, and show that there is agreement with our previous results on the complete data subset (section "Improved fine-mapping of signals in latent factor GWAS estimated from summary statistics"). To show the generalisability of our approaches, we include an application to nearly 200 NMR metabolic traits, where we estimate latent factor GWAS summary statistics and fine-map several regions (section "Latent factors underlying variation in metabolic traits provide improved fine-mapping resolution").

- The study only retains a portion of the dataset for efficiency reasons, raising questions about the required data volume. A systematic benchmarking analysis should be conducted to identify the minimum sample size necessary for maintaining fine-mapping accuracy. This would provide valuable guidance for future users of flashfmZero on the minimum dataset thresholds.

Thanks to the helpful comment of reviewer 2, we no longer need to down-sample to obtain complete data. We have derived latent factor GWAS effect estimates that only require the factor loading matrix (obtained from a trait correlation matrix) and observed trait GWAS effect estimates, which may be from the full sample where some individuals do not have some trait measurements. See our new subsection "GWAS of latent factors using summary statistics" in the Methods details section. This new approach for latent GWAS estimation using summary statistics is also provided in our flashfmZero R package.

- To demonstrate the superiority of flashfmZero, the authors should compare its performance with alternative methods, such as mvSuSiE (an extension of SuSiE, a preprint, PMID37425935), CAFEH (PMID35085493), or other multi-trait fine-mapping approaches. Using both real and simulated datasets would give a more comprehensive evaluation of flashfmZero's advantages and limitations. *It is difficult to compare methods in real data where the underlying causal variants are unknown, and we have previously published simulation comparisons for flashfm and mvSuSiE, where traits are correlated and applied within each of European and African ancestries. As flashfmZero makes use of the previously published flashfm framework, we refer to our previously published simulation comparisons of mvSuSiE and flashfm – there we showed that flashfm has slightly higher power and lower false discovery rate (FDR) than mvSuSiE, and that mvSuSiE has noticeably higher FDR in the*

European sample (<https://www.nature.com/articles/s41467-023-43159-5/figures/4>). We have added the following to the fine-mapping section within Results:

"In previous simulation comparisons of fine-mapping associations for correlated traits within European ancestry individuals and within African ancestry individuals, flashfm was found to have slightly higher power and lower false discovery rate (FDR) than mvSuSiE (Zhou et al. 2023). Moreover, mvSuSiE had noticeably higher FDR in the European sample, where there are longer linkage disequilibrium (LD) blocks. For this reason, we focus on flashfmZero for multi-trait fine-mapping of uncorrelated latent factors."

Minor Comments:

1. Terminology and abbreviations

In the results section, blood trait abbreviations (e.g., RBC-SFL-DW) should be directly defined in the text, rather than referring readers to supplementary tables. This would enhance accessibility and prevent unnecessary interruptions for readers unfamiliar with the acronyms.

Thank you for this comment that improves accessibility to blood cell trait abbreviations. We have included the long names of each red blood cell trait after the first instance of each abbreviation in a paragraph, e.g. "RBC-SFL-DW (red blood cell side fluorescence distribution width)"

2. Statistical descriptions

The results section should include more interpretation of the descriptive statistics presented. For example, statements like "76% of the comparisons" and "strictly smaller in 58% of comparisons" need more explanation to clarify what constitutes a "strict" comparison and how these percentages are defined. Including thresholds or clarifying metrics will make the statistical claims easier to understand.

Thank you for this suggestion which improves the presentation of these results. We have re-written the comparative paragraph and expanded on it to provide more context:

"To compare the results of the fine mapping of the blood cell traits (single-trait fine-mapping) to those of fine-mapping of the latent factors (single-trait fine-mapping and multi-trait fine-mapping), we matched each blood cell trait to the latent factor that is the highest contributor to it, amongst latent factors that had a signal in the corresponding region (Figure S4). This resulted in 1,238 comparisons of 99% credible set (CS99) sizes (number of variants in the CS99) between 'JAM blood cell trait' and 'JAM latent factor'. In regions that contained association signals with at least two latent traits, we also compared the results of the fine-mapping of the blood cell traits and latent factors with 'flashfm latent factor', which provides a CS99 for each latent factor. This resulted in 725 comparisons of CS99 sizes between 'JAM blood cell trait' and 'flashfm latent factor' and 211 comparisons of CS99 sizes between 'JAM latent factor' and 'flashfm latent factor'. In the comparisons of 'JAM blood cell trait' with 'Flashfm latent factor', we followed the same procedure used for 'JAM blood cell trait' and 'JAM latent factor' comparisons, i.e. matched each blood cell trait to the latent factor that is the highest contributor to it, amongst latent factors that had a signal in the region. For comparisons between 'JAM latent factor' and 'flashfm latent factor' we matched by latent factor as these methods both return a CS99 for each latent factor.

Let $CS99_{JAM\ blood\ cell\ trait}$ be the number of variants in a 'JAM blood cell trait' CS99, let $CS99_{JAM\ latent\ factor}$ be the number of variants in a 'JAM latent factor' CS99 and let $CS99_{flashfm\ latent\ factor}$ be the number of variants in a 'flashfm latent factor' CS99. We refer to a method as having improved resolution over another method if it tends to construct smaller CS99s than the other method.

`JAM latent factor' has improved resolution over `JAM blood cell trait'. In 76% (937/1238) of the `JAM blood cell trait' vs `JAM latent factor' comparisons, we had $CS99_{JAM\ latent\ factor} \leq CS99_{JAM\ blood\ cell\ trait}$; in 58% (725/1238) of these comparisons we had $CS99_{JAM\ latent\ factor} < CS99_{JAM\ blood\ cell\ trait}$ (Figure 5, Table S8). `flashfm latent factor' further improves resolution over `JAM blood cell trait', as $CS99_{flashfm\ latent\ factor} \leq CS99_{JAM\ blood\ cell\ trait}$ in 86% (624/725) of the comparisons and $CS99_{flashfm\ latent\ factor} < CS99_{JAM\ blood\ cell\ trait}$ in 71% (517/725) of the comparisons. When latent factors have no shared causal variants, as suggested by no overlap in their CS99, `flashfm latent factor' and `JAM latent factor' will give similar results (as previously illustrated for flashfm applied to correlated traits[4]). However, flashfm will have improved resolution over single-trait methods when there are shared causal variants amongst the traits. Consequently, we observe that $CS99_{flashfm\ latent\ factor} \leq CS99_{JAM\ latent\ factor}$ in 97% (205/211) of the comparisons and $CS99_{flashfm\ latent\ factor} < CS99_{JAM\ latent\ factor}$ in 45% (95/211) of the comparisons (Figure 5, Table S8)."

3. Improving figures and visual aids

- Figure 1: The abbreviations used (e.g., IG%) should be clarified in the figure legend to ensure that readers without specific knowledge of blood cell traits can understand the terms used.

We have added text to the beginning of the caption for Fig 1 to highlight Table S1 for details on blood cell traits:

"Descriptions of the blood cell traits, including their standard abbreviations and full names, are given in Table S1."

- Figure 4: The axes need clearer labeling. Providing detailed explanations for the x-axis and y-axis would improve readers' ability to quickly interpret the data presented in the plot.

We have modified Figure 4 so that each axis includes "CS99 size", e.g. "JAM latent factor" has been changed to "JAM latent factor CS99 size".

- Figure 6: The distinction between SNP groups should be explained more thoroughly. The use of letters (A-C) versus numbers (0) should be justified or clarified to help readers understand the coding system applied to these SNP groups.

Thank you for this good point and we have clarified this in the caption:

"The colours of the circles (labelled by letters in the legend) indicate groups of variants (with MPP > 0.01) in high LD ($r^2 > 0.8$) as calculated by the fine-mapping method; variants labelled as group "0" had MPP < 0.01 and were not assigned to a group."

4. Methods

The methods section mentions selecting the optimal number of latent factors (i.e., 25), but lacks a detailed description of how this was determined. Supplementary figures illustrating the process for choosing the optimal number of latent factors would make the methodology more transparent.

We now include Figure S1 which shows the scree plot that we used to select 25 latent factors. We have also included more detail on this in the Method section, as detailed in Major Comment 2, above. We also provide scree plots for our two new analyses using our summary-statistic based approach for blood cell traits (Figure S9) and for NMR traits (Figure S11).

5. R Packages and code accessibility

The flashfmZero package has installation issues on MacOS, specifically relating to the "gfortran" library. The authors should provide a troubleshooting guide for users who encounter this problem, along with any relevant resources to ensure smooth installation across operating systems.

Additionally, some users have reported errors when running showcase examples on the flashfmZero

website (e.g., "maxcv_stop" errors). Ensuring the reproducibility of these examples and providing clear input data formats on GitHub would increase the accessibility and usability of the method for a broader audience.

Thank you for this helpful comment that will increase accessibility and usability. We have updated the installation instructions on GitHub and provide clearer input data requirements. The maxcv_stop issue is a closed/resolved issue (in 2023) from the published MGflashfm software and is not an issue in flashfmZero.

*We have **updated our installation instructions** on GitHub, giving details for Mac and Windows separately, and stating availability for all platforms. For Mac use, the standard tools of Xcode and a Fortran compiler are required. Our instructions are now as follows:*

"flashfmZero could be installed with ease on versions of R > 4.2.1 and is compatible with all platforms.

Installation time is estimated as 2 minutes.

Specific requirements for Windows and Mac platforms follow.

Windows

Must install [Rtools](<https://cran.r-project.org/bin/windows/Rtools/>).

Mac

Must have the following installed (details at [R for MacOS](<https://cran.r-project.org/bin/macosx/tools/>)):

- 1. Xcode: free on the Apple App Store*
- 2. Fortran compiler. R 4.3.0 and higher use universal GNU Fortran 12.2 compiler and an installer package is available here: [gfortran-12.2-universal.pkg](<https://mac.r-project.org/tools/gfortran-12.2-universal.pkg>) (242MB)*

*We have given an **overview of the required data input** (with reference to detailed data examples, which are provided in the package) before providing the available functions for fine-mapping:*

****Required Data****

[Example data](#example-data) are described in detail below. Briefly, all multi-trait functions require:

1. [gwas.list](#smim1-gwas-list-interval-fa25): a list of data.frames (one for each trait), with required (minimal) columns:

(a) ****rsID****: (variant ID names - if using the JAM functions, variant names must not contain ":" and must start with a non-numeric character, see [Notes](#notes) for details). All rsIDs must be unique, i.e. no duplicate names.

(b) ****beta****: GWAS effect sizes

(c) ****EAF****: effect allele frequency; frequency of effect allele that coincides with beta

2. [corX](#smim1-corx): SNP correlation matrix where row and column names are the same variant ID names as in gwas.list, and appear in the same order as in gwas.list. ****NOTE****: The effect allele coding for beta MUST match with the allele coding in corX; either all effect alleles in the GWAS must match with the reference allele in corX or all non-effect alleles must match with the reference allele

in corX (need consistency for correlation signs). Where there is a mis-match in alleles between gwas.list and corX, flip the variant in gwas.list by setting beta to -beta and EAF to 1-EAF at that variant.

To assist in this we provide the ****alignGWAS**** function.

3. [N](#smim1-N): a single number that gives the sample size of the cohort

The single-trait functions require the same input, except the GWAS argument is **gwas**, a data.frame as in gwas.list, i.e. gwas has the form of gwas.list[[1]].

We have also included an example dataset to run factor analysis and latent factor GWAS estimation using our new summary-statistic based approach.

6. Discussion and conclusion

The current discussion does not explore the potential for applying flashfmZero to other categories of high-dimensional traits or disease models. Including a discussion of the tool's applicability to other biological domains, such as rare diseases, or its utility in drug target prioritization (such as Open Targets Genetics / PMID34711957 and Priority index / PMID31253980) would broaden the impact of the research and open avenues for future exploration.

Thank you for this helpful comment to expand the Discussion. We have explored applications to binary traits (including rare diseases) and utility in drug target prioritisation.

Considering your comment and comment 5 of Reviewer 2, we have added the following on application to binary traits and the possibility for rare diseases:

“We have demonstrated the use of flashfmZero on uncorrelated latent factors that were derived from quantitative traits, where the factor loading matrix is obtained either from the individual-level quantitative traits data or their trait correlation matrix. Similar to flashfm[4], we may apply our latent factor framework and flashfmZero to binary traits by using the genetic correlation as estimated by cross-trait LD Score regression (Bulik-Sullivan et al. 2015). Binary trait GWAS summary statistics that have been estimated using a linear model may be used directly in flashfm, whereas effect estimates from a logistic model should be converted to a linear approximation (Cook et al. 2017). If the binary traits have a low/zero correlation, then factor analysis is inappropriate, but flashfmZero multi-trait fine-mapping could be applied directly to any number of binary traits, provided that the GWAS summary statistics are on a linear scale. For correlated binary traits, factor analysis may be applied by using the genetic correlation to estimate the number of latent factors that underpin multiple outcomes and the corresponding factor loadings. Then, latent factor GWAS summary statistics may be calculated using the “latentGWAS” function in our flashfmZero R package, as in the flow depicted in Figure 2(B), followed by fine-mapping with flashfmZero (Figure S4). Further work is needed to understand possible applications to rare diseases, as to how well latent factors could explain their variability.”

We have added the following on drug target prioritisation:

“Finally, there is potential for flashfmZero to inform shared therapeutics, opening avenues for future exploration. For example, a recent pleiotropy-driven approach, PIPE (Pleiotropy Informing Prioritization and Evaluation) was developed to focus on pleiotropic evidence in the prioritisation and evaluation of therapeutic targets (Bao et al. 2024). PIPE considers genetic variants that have been identified by cross-disease GWAS, and its authors discuss potential further improvements by including potential shared causal variants, as from flashfm multi-trait fine-mapping results. Integration of latent factors underlying many diseases or disease-related quantitative traits, and the improved prioritisation of their shared causal variants via flashfmZero, may reveal additional insights that would benefit PIPE and enhance contributions to shared therapeutics. Priority index (Fang et al.

2019) and Open Targets Genetics(Mountjoy et al. 2021) are other popular approaches for translating GWAS associations into drug target prediction, and do not yet integrate pleiotropic evidence. However, these two approaches incorporate colocalisation results between trait GWAS and expression quantitative trait loci (eQTL). Considering the causal variant prioritisation gains (i.e. high marginal posterior probability of causality) attained with flashfmZero, it has high potential to identify shared causal variants underlying latent factors that explain multiple diseases and traits. Moreover, leveraging information between latent factors gives an alternative strategy to gain power that is crucial for smaller GWAS, as is common in eQTL studies and in under-represented populations. Integration of flashfmZero with Priority index and Open Targets Genetics is a future avenue for improvement in drug target identification and prioritisation.”

Reviewer #2: The authors present flashfmZero, a novel methodology for fine-mapping in genome-wide association studies (GWAS) of high-dimensional traits. This approach builds upon the existing flashfm method and offers significant advantages in elucidating GWAS findings, particularly in the context of correlated traits and multi-morbidities. By leveraging latent factors derived from multiple related traits, flashfmZero aims to enhance both discovery power and fine-mapping precision.

1. The method's use of latent factors to capture shared biological mechanisms across multiple traits is innovative and potentially powerful.
2. The results suggest that flashfmZero can produce more precise credible sets compared to single-trait analyses, which could lead to more efficient identification of causal variants.
3. The ability to detect signals that are not genome wide significant with several traits and are minimally associated demonstrates the method's potential to uncover previously undetected genetic associations.

Here are some points for concerns/clarification:

1. Since the method can only be applied in the individual level datasets, it presents with a big challenge in the way GWAS is conducted in modern times, i.e. meta-analyzing data from multiple resources and then finamapping. Authors should present with some future directions and possibilities on how this method can be make applicable to non-individual level datasets. Potential avenues could include developing summary-statistic based approximations or potentially exploring federated learning approaches to maintain privacy while leveraging multi-cohort data.

We thank the reviewer for this helpful comment. We have derived an expression for the latent factor GWAS effect estimates that only requires the factor loading matrix (obtained from the trait correlation matrix) and the observed traits' GWAS effect estimates. We have added a comment on this to the section "Factor analysis of quantitative traits":

"We later relax the requirement of complete data by deriving an estimate of latent factor GWAS effect estimates that only requires the factor loading matrix (obtained from factor analysis using the trait correlation matrix) and observed trait GWAS effect estimates. This avoids the need for individual-level data and allows flexibility to missing trait measurements."

Within the Introduction we have added:

"In addition, we show that latent factor GWAS summary statistics can be derived from the GWAS summary statistics of the observed traits and a factor loading matrix computed from the trait correlation matrix. This approach extends the applicability of flashfmZero to summary-level datasets, offering the flexibility to include individuals with incomplete trait measurements."

This change is reflected in our Discussion by adding:

"We first focused on the estimation of latent factor GWAS effect estimates from first principles, where individual-level data were available on the traits and genotypes. The subset of study participants who had no missing trait measurements (i.e. complete data) was required to enable calculation of their latent factor scores. Then, the genotypes were needed to perform the genetic association analysis for the latent factors (latent factor scores) and to estimate LD for fine-mapping.

The latent factor GWAS using individual-level data is performed independently of the observed trait GWAS. However, we also derived an approach to estimating latent factor GWAS using only summary-level data - the observed trait GWAS summary statistics and the covariance matrix of the observed traits. This estimation removes the need for data where all individuals have measurements for all traits. In addition, it widens the use of our approach to summary-level datasets freely available online, where the trait correlation may be estimated directly from the traits data or by using methods such as cross-trait LD Score regression(Bulik-Sullivan et al. 2015). We note that estimation of the latent factor GWAS summary statistics from the observed trait GWAS summary statistics requires that the variant is present for all observed traits. However, as the traits are measured in the same cohort, the majority of variants are expected to be present for all traits."

2. The paper lacks discussion on how flashfmZero performs across different genetic ancestry groups. Authors should address how the method accounts for population structure and whether it can be applied to or interpreted for multi-ancestry studies.

Thank you for this interesting comment. Extension to multi-ancestry studies is not straightforward and is future work. We have added the following to the Discussion to address this.

“Our current implementation is for a single genetic ancestry group, and we account for population structure by using GWAS summary statistics from a mixed linear model (e.g. BOLT-LMM(Loh et al. 2015)). Extension to multi-ancestry studies would be a useful future extension, although this is not straightforward. It is possible to conduct latent factor GWAS within each genetic ancestry group. However, care would be needed to ensure that the latent factors from each group could be mapped between groups and are representing similar observed traits. Then, the same latent factor may be analysed across multiple ancestries following approaches such as MR-MEGA(Mägi et al. 2017) for single latent factor meta-analysis of GWAS, whilst fine-mapping signals in multiple latent factors across multiple genetic ancestries may be achieved using MGflashfm(Zhou et al. 2023) or an extension of MeSuSiE(Gao and Zhou 2024).”

3. Comparisons with ancestry-specific fine-mapping methods like MR-MEGA or extensions of MeSuSiE for multi-trait analysis could provide valuable insights.

We agree that ancestry-specific fine-mapping methods are another strategy to improve fine-mapping resolution. However, as described in the above response, it is not straightforward to extend our latent factor approach to multiple ancestries. Above, we discuss possible extensions involving MR-MEGA, MGflashfm, and extensions of MeSuSiE.

4. Exploring the use of ancestry-specific LD panels within the flashfmZero framework could enhance its applicability to diverse populations.

We agree and have added to the Discussion that adapting flashfmZero to include multiple ancestry-specific LD panels is future work, which is not straightforward:

“MGflashfm allows for the inclusion of genetic variants that are not present in all ancestry groups, so that causal variants that are shared amongst multiple ancestries or ancestry-specific are identifiable. MGflashfm for multi-group/ancestry multi-trait fine-mapping makes use of flashfm multi-trait fine-mapping to leverage information between traits within each genetic ancestry group and joint

analysis across multiple groups using ancestry-specific LD panels. Future work includes adapting our latent factor GWAS framework and flashfmZero to the MGflashfm framework for multi-group/ancestry and multiple latent factor fine-mapping.”

5. The method is only applied on quantitative traits, is it applicable for binary traits. Please elaborate.

This is a good point, and we have clarified in the Discussion, that as with flashfm, our latent factor approach and flashfmZero are applicable to binary traits:

“We have demonstrated the use of flashfmZero on uncorrelated latent factors that were derived from quantitative traits, where the factor loading matrix is obtained either from the individual-level quantitative traits data or their trait correlation matrix. Similar to flashfm[4], we may apply our latent factor framework and flashfmZero to binary traits by using the genetic correlation as estimated by cross-trait LD Score regression(Bulik-Sullivan et al. 2015). Binary trait GWAS summary statistics that have been estimated using a linear model may be used directly in flashfm, whereas effect estimates from a logistic model should be converted to a linear approximation(Cook et al. 2017). If the binary traits have a low/zero correlation, then factor analysis is inappropriate, but flashfmZero multi-trait fine-mapping could be applied directly to any number of binary traits, provided that the GWAS summary statistics are on a linear scale. For correlated binary traits, factor analysis may be applied by using the genetic correlation to estimate the number of latent factors that underpin multiple outcomes and the corresponding factor loadings. Then, latent factor GWAS summary statistics may be calculated using the “latentGWAS” function in our flashfmZero R package, as in the flow depicted in Figure 2(B), followed by fine-mapping with flashfmZero (Figure S4). Further work is needed to understand possible applications to rare diseases, as to how well latent factors could explain their variability.”

6. Its hard to understand how latent factors can be constructed for a new dataset. I looked through the code in Github and it only shows the application from this paper. To make this tool more broadly available, it is important to expand the Github to explain its applicability to broader questions.

We agree that a more general script would be helpful to guide users in other analyses. We have provided general scripts as articles on the flashfmZero GitHub page. They are fully annotated for our NMR analysis and easily generalised to other data; this same script (with minor changes) was used for our summary-statistic-based analysis of the blood cell trait data. (See https://jennasimit.github.io/flashfmZero/articles/Example_Part1.html for “Estimation of latent factors” and https://jennasimit.github.io/flashfmZero/articles/Example_Part2.html for “Latent factor GWAS and fine-mapping”).

In our Code Availability statement we have added “Fully annotated scripts for the summary-statistics based approach applied to the NMR traits of INTERVAL are available as articles at <https://jennasimit.github.io/flashfmZero>[34]. This same code was used for the summary-statistic based analysis of the blood cell traits, but with minor changes.”

We have also provided new automated functions to assist with analysis - such as “factor_contributions” to assist with interpretations and “alignGWAS” to harmonise the GWAS datasets to the same effect allele and to the coding used in the LD matrix, as well as the wrapper “harmoniseGWAS” to harmonise all GWAS datasets. The wrapper function `FMsummary_table_general` summarises fine-mapping results over the many traits and latent factors, constructing tables similar to our Tables S9 and S14.

We also provide example data to run factor analysis and estimate latent GWAS summary statistics using our new summary-statistic based method (see <https://jennasimit.github.io/flashfmZero/articles/flashfmZero.html>).

7. Interpretation of latent factor results is also difficult, Authors should offer guidelines on interpreting results and selecting appropriate parameters.

Thank you for this helpful point (also see our response to major comment 2 of Reviewer 1). We have added a new section to clarify this aspect, as well as a new function (factor_contributions) in our flashfmZero package, which calculates the latent factor contributions from the factor loading matrix and outputs the matrices in an easier-to-interpret form, grouping observed traits by highest contributing latent factor.

Our new section within Method Details:

“Interpretation of latent factors

Let L_{ij} be the factor loading of latent factor j ($j=1,\dots,L$) for raw trait i ($i=1,\dots,P$). We define the contribution of latent factor j to raw trait i by $C_{ij} = \frac{L_{ij}^2}{\sum_{k=1}^L L_{ik}^2}$, to aid in mapping the contributions of the latent factors back to each raw trait. These scaled factor loadings indicate the proportion of variance in each observed trait i that is explained by latent factor j , relative to the total variance explained jointly by the latent factors. That is, for each observed trait, the contributions from all factor loadings sum to one.

To understand which observed traits are explained by each latent factor, we collect observed traits that have the same top-contributing latent factor. We automate this in our “factor_contributions” function within the flashfmZero package[34], which takes the factor loading matrix as input and returns the latent factor contributions (re-scaled factor loadings) and factor loading matrix with observed traits ordered by maximum contributing latent factor.”

Referees' report, second round of review

Reviewer 1

Overall, almost all of my concerns have been satisfactorily addressed. However, there are still a couple of minor points for potential refinement.

Firstly, regarding the determination of latent factors, while authors have explained well about their number and biological significance, there is a bit of a discrepancy. It has been mentioned using the Horn parallel method to identify

25 latent factors, yet Figure S1 presents a scree plot of eigenvalues. These are two distinct approaches for factor determination, and the current text description does not align with the figure. It would be beneficial to provide a more precise and detailed account of the methodology employed to ascertain the number of factors, ensuring consistency between the written text and visual representation.

Secondly, a comparison of the performance of flashfmZero, mvSuSiE, and CAFEH on the same dataset would have added significant value and provided a more comprehensive understanding of the proposed approach's superiority or uniqueness. This omission may leave the reader with some lingering questions.

While these are not major issues, addressing them would enhance the overall quality and clarity of the manuscript.

Reviewer 2

Authors did a great job at addressing all comments thoroughly and addressing limitations as well.

Authors' response to the second round of review

Firstly, regarding the determination of latent factors, while authors have explained well about their number and biological significance, there is a bit of a discrepancy. It has been mentioned using the Horn parallel method to identify 25 latent factors, yet Figure S1 presents a scree plot of eigenvalues. These are two distinct approaches for factor determination, and the current text description does not align with the figure. It would be beneficial to provide a more precise and detailed account of the methodology employed to ascertain the number of factors, ensuring consistency between the written text and visual representation.

Thanks for noticing this discrepancy. We have added additional details to our related Methods section:

“The number of latent factors is selected based on the observed trait data, and this decision is often made using Horn’s parallel method. In Horn’s parallel methods, eigenvalues are calculated from the observed data and from “noisy” random data. These two sets of eigenvalues are often compared in a scree plot, which displays the eigenvalue for each number of factors. The eigenvalues of the observed data will be larger than those from the random data until a certain point - this point where the observed data eigenvalues first become smaller than those from random data is the suggested number of factors.”

“Based on Horn’s method, a scree plot (using “fa.parallel()” in the psych package) that compares the eigenvalues calculated in the data and in random datasets indicated that 25 latent factors was an optimal choice (Figure S1); the number of latent factors is selected such that the data-calculated eigenvalues are larger than those based on the random datasets.”

We have updated the text in Fig 2 to include “Horn’s parallel method” and we have updated the three relevant supplementary figures (Figures S1, S9, S11) to include:

“..., as suggested by Horn’s parallel method implemented in the “psych::fa.parallel” function in R,...The blue line indicates the eigenvalues calculated in the observed data and the red dashed line corresponds to those calculated in random (noisy) data sets. The selection of 25

We thank the reviewers for their constructive comments which have strengthened our manuscript. We have clarified the latent factor selection and have included comparisons of flashfmZero with mvSuSiE, focusing on two blood cell trait regions in our analysis of the INTERVAL study.

In particular, we found that in smaller studies (~20,000 individuals), mvSuSiE was unable to dissect the relationships between the variants and the traits, but was able to do so in a substantially larger study (>200,000 individuals). However, flashfmZero applied to the smaller study showed agreement with mvSuSiE applied to the larger study.

Reviewers' Comments:

Reviewer #1: Overall, almost all of my concerns have been satisfactorily addressed. However, there are still a couple of minor points for potential refinement.

We thank the reviewer for their insights in improving this work and detail our changes to address the remaining points below.

factors is the point where the observed data eigenvalues are smaller than those in the random data." (for Figs S1, S9, and appropriately adjusted for S11).

Secondly, a comparison of the performance of flashfmZero, mvSuSiE, and CAFEH on the same dataset would have added significant value and provided a more comprehensive understanding of the proposed approach's superiority or uniqueness. This omission may leave the reader with some lingering questions.

We have included comparisons between flashfmZero and mvSuSiE for blood cell trait fine-mapping in two regions that we have previously highlighted: SMIM1 and PIEZO1. We were unable to get CAFEH to run successfully on this scale of dataset.

We have added the following within our Results section and present details in a new Table S9:

"As mvSuSiE requires complete data[13], we applied it within the subset of 18,310 individuals and compared the results with those from flashfmZero in regions highlighted as having biologically likely causal variants. Our summary statistics version of flashfmZero can further improve fine-mapping resolution over methods requiring complete data because it is possible to include individuals who do not have measurements for all traits."

...

“After mvSuSiE identifies a variant or variants with high posterior probability of causality for at least one trait (PIP), its local false sign rate (lfsr) is used to interpret which traits are associated with the variant(s); a recommended threshold is $\text{lfsr} < 0.01$ [13]. The application of mvSuSiE to all the traits observed to have a signal in the *PIEZO1* region also identified two potential causal variants: rs551118 (16:88856084) and rs904801 (16:88517105) with PIP 0.978 and 0.945, respectively. However, none of the traits had $\text{lfsr} < 0.01$ at either of these variants (Table S9). Weakening the threshold to $\text{lfsr} < 0.05$, suggests that all traits (related to red blood cells and basophils) have rs551118 as a causal variant, and none are impacted by rs904801. However, rs551118 is unlikely to be causal for basophil-related traits which have p-values between 9.9×10^{-4} and 0.038, whereas rs904801 is genome-wide significant for these traits ($P < 2.4 \times 10^{-10}$) (Table S5). Moreover, in an application of mvSuSiE to 16 blood cell traits within the complete data of the European ancestry subset of UK Biobank (248,980 individuals), rs551118 was identified as a causal variant for red blood cell traits, but not basophils[13]. This suggests that large samples (e.g. biobank size) are needed for mvSuSiE to identify accurately which traits have particular causal variants. For smaller studies (~20,000 individuals), mvSuSiE contributes to high-level identification of potential causal variants, but does not to a refined interpretation of the traits that are impacted by particular variants.”

...

“The application of mvSuSiE to the *SMIM1* region gave results that agreed with flashfmZero, identifying rs1175550 (1:3691528) as a causal variant for at least one trait (PIP=0.999). All traits with a signal in the region had $\text{lfsr} < 0.01$ (Table S9). This suggests that in samples of this size (~18,000 individuals) mvSuSiE performs well when all traits share a causal variant,

but that it has some difficulty in identifying the traits impacted by causal variants when a region contains distinct causal variants for subsets of traits (as for *PIEZO1*). ”

And the following to our Discussion:

“We also compared flashfmZero with an alternative multi-trait fine-mapping approach, mvSuSiE[13], which requires complete data. In a region (*SMIM1*) where all traits showed evidence for association with the same causal variant, flashfmZero and mvSuSiE showed agreement. However, in another region (*PIEZO1*) in which flashfmZero exhibited evidence for distinct causal variants for subsets of different traits, mvSuSiE was unable to dissect the relationships between the variants and the traits in the INTERVAL study complete data subset of 18,310 individuals. In a substantially larger study of more than 200,000 individuals[13], mvSuSiE was able to differentiate the association signals in this region in agreement with the results of flashfmZero applied to the sample of ~18,000 (and ~43,000) individuals.”

And the following to our Methods:

“For comparison purposes, we applied mvSuSiE[13] to two regions where the likely causal variant has biological support. In our implementation of mvSuSiE we used the canonical prior and followed the author’s suggestion of estimating the residual variance using the variants with absolute Z score below 2 for all traits; we also set coverage to 0.99.”

While these are not major issues, addressing them would enhance the overall quality and clarity of the manuscript.

Reviewer #2: Authors did a great job at addressing all comments thoroughly and addressing limitations as well.

We thank the reviewer for their insights in improving this work.